# Learnability with Indirect Supervision Signals

**Kaifu Wang**
University of Pennsylvania
kaifu@sas.upenn.edu

**Qiang Ning**[*]
Amazon
qning@amazon.com

**Dan Roth**
University of Pennsylvania
danroth@seas.upenn.edu

## Abstract

Learning from indirect supervision signals is important in real-world AI applications when, often, gold labels are missing or too costly. In this paper, we develop a unified theoretical framework for multi-class classification when the supervision is provided by a variable that contains nonzero mutual information with the gold label. The nature of this problem is determined by (i) the transition probability from the gold labels to the indirect supervision variables and (ii) the learner's prior knowledge about the transition. Our framework relaxes assumptions made in the literature, and supports learning with unknown, non-invertible and instance-dependent transitions. Our theory introduces a novel concept called *separation*, which characterizes the learnability and generalization bounds. We also demonstrate the application of our framework via concrete novel results in a variety of learning scenarios such as learning with superset annotations and joint supervision signals.

## 1 Introduction

We are interested in the problem of multiclass classification where direct and gold annotations for the unlabeled instance are expensive or inaccessible, and instead the observation of a dependent variable of the true label is used as supervision signal. Examples include learning from noisy annotations [1, 21, 26], partial annotations [16, 22, 14] or feedback from an external world [15, 8].

To extract the information contained in a dependent variable, the learner should have certain *prior knowledge* about the *relation* between the true label and the supervision signal, which can be expressed in various forms. For example, in the noisy label problem, the noisy rate is assumed to be bounded by a constant (such as the *Massart noise* [23, 17]). In the superset problem, the true label is commonly assumed to be contained in (or consistent with) the superset annotation [16, 22].

As in [13, 29, 36], we model the aforementioned relation using a *transition probability*, which is the distribution of the observable variable conditioned on the label and instance. The transition enables the learner to induce a prediction of the observable via the prediction of the label, and construct loss functions based on the induced prediction and the observable.

In this paper, instead of assuming that the learner *fully knows* the transition, we formalize the concept of *transition class*, a set that contains all the candidate transitions, to describe more general forms of prior information. Also, we define the concept of *separation* to quantify whether the information is enough to distinguish different labels. With these concepts, we are able to study a variety of learning scenarios with unknown, non-invertible and instance-dependent transitions in a unified way. We show this under the *realizability assumption* (also called *separable* in linear classification), a commonly made assumption (such as [2, 19, 22]) that assumes that the true classifier is in the hypothesis space.

Our goal is to develop a *unified theoretical framework* that can (i) provide learnability conditions for general indirect supervision problems, (ii) describe what prior knowledge is needed about the transition, and (iii) characterize the difficulty of learning with indirect supervision.

---

[*]Work done while at the Allen Institute for AI and at the University of Illinois at Urbana-Champaign.

Specifically, in this paper, our main contribution includes:

1. We decompose the learnability condition of a general indirect supervision problem into three aspects: *complexity*, *consistency* and *identifiability* and provide a unified learning bound for the problem (Theorem 4.2).

2. We propose a simple yet powerful concept called *separation*, which encodes the prior knowledge about the transition using statistical distance between distributions over the annotation space and uses it to characterize *consistency* and *identifiability* (Theorem 5.2).

3. We formalize two ways to achieve separation: total variation and joint supervision, and use them to derive concrete novel results of practical learning problems of interest (Section 5.2 and 5.3).

All proofs of the theoretical results are presented in the supplementary material.

## 2 Related Work

**Specific Indirect Supervision Problems.** Our work is motivated by many previous studies on the problem of learning in the absence of gold labels. Specially, the problem of classification under *label noise* dates back to [1] and has been studied extensively over the past decades. Our work is mostly related to (i) Theoretical analysis of PAC guarantees and consistency of loss functions, including learning with bounded noise [23, 21, 2], and instance-dependent noise [30, 24, 12]. (ii) Algorithms for learning from noisy labels, including using the inverse information of the transition [26, 36], and inducing predictions of noisy label (which is more similar to our formulation) [9, 34].

*Superset* (also called *partial label*) problems, where the annotation is given as a subset of the annotation space, arises in various forms in standard multiclass classification and structured prediction [16, 14, 20, 27]. While it is possible to extend some approaches in the theory of noisy problems to the superset case, the superset problem focuses on the case of a large and complex annotation space, and some of the assumptions (such as "known transition") would be too strong in practice. On the theoretical side, [16] defines *ambiguity degree* to characterize the learning bound. [22] provides an insightful discussion of the PAC-learnability of the superset problem and proposes the concept of *induced hypothesis*. This two papers motivate the approach pursued in this paper.

**Frameworks for Indirect Supervision.** Our supervision scheme is conceptually similar to [33, 29], which model the label as a latent variable of the indirect supervision signal. However, the discussion is restricted to the exponential family model. [10, 11] also propose a framework and algorithms to supervise a structured learning problem with an indirect supervision, which is modeled as a binary random variable associated with the gold label. Our work extends the binary indirect signal to a multiclass signal and gives a theoretical treatment to this general learning problem. [13, 14] study the problem of designing consistent loss functions for superset problems when the transition (aka *mixing*) matrix is *partially known*. The discussion can be applied to a wider range of problems such as noisy and semi-supervised learning. Our framework can also be compared to the multitask learning framework proposed in [5, 6], which defines a notion called the $\mathcal{F}$-relatedness to describe the relation machine learning tasks through some deterministic functions within the instance space $\mathcal{X}$. As contrast, our framework studies the probabilistic transitions between different domains (from the label space $\mathcal{Y}$ to the annotation space $\mathcal{O}$) and our gold label $Y$ is not observable, which is different than a multitask setting. Our goal is mostly related to [36], which further develops the ideas from [32, 14, 26] and develops a general framework of learning from data with *reconstructible* corruption, using the inverse of a known, instance-independent transition matrix, to construct unbiased estimator of the classification loss and derive generalization bounds. Our study aims to relax these assumption on the transition matrix, especially when the label space and/or annotation space is large and the estimation of the transition matrix could be difficult.

## 3 Preliminaries

We will use $\mathbb{P}(\cdot)$ to denote probability, $\mathbb{E}[\cdot]$ to denote expectation, $\mathbb{1}\{\cdot\}$ to denote the indicator function and $p(\cdot)$ to denote the density function or more generally, the Radon–Nikodym derivative.

We denote the source variable as $X$, which takes value in an input space $\mathcal{X}$ and denote the target label as $Y$, which takes value in a label space $\mathcal{Y}$. We assume $|\mathcal{Y}| = c$ is finite and identify the elements in

$\mathcal{Y}$ as $\{y_1, y_2, \ldots, y_c\}$. The goal is to learn a mapping $h_0 : \mathcal{X} \to \mathcal{Y}$. The *hypothesis class* $\mathcal{H}$ contains candidate mappings $h : \mathcal{X} \mapsto \mathcal{Y}$. The loss function for hypothesis $h$ and sample $(x, y)$ is denoted as $\ell(h(x), y)$. The *risk* of a hypothesis $h \in \mathcal{H}$ is defined as $R(h) := \mathbb{E}_{X,Y}[\ell(h(x), y)]$, where $x$ is sampled independently from a (unknown) distribution $D_X$. We will focus on the realizable case, i.e., there is a classifier $h_0 \in \mathcal{H}$ such that $R(h) = 0$. As in standard PAC-learning theory, we use the zero-one loss for the gold sample $(x, y)$ (although we may not observe $y$): $\ell(h(x), y)) = \mathbb{1}\{h(x) \neq y\}$.

An *annotation* $O$ (also called *supervision signal*) is a random variable that is not independent with $Y$ (or equivalently, $O$ and $Y$ has positive mutual information). The dependence between $X$ and $O$ conditioned on $Y$ is allowed but not required. $O$ takes value in an annotation space denoted as $\mathcal{O}$. We also assume $|\mathcal{O}| = s < \infty$ and identify the elements in $\mathcal{O}$ as $\{o_1, o_2, \ldots, o_s\}$. For convenience, when using $y_i$ and $o_i$ as subscripts, we regard $y_i, o_i$ as its index $i$. For example, for any indexed quantity $a_i$, we will denote $a_{y_i}$ as $a_i$. We denote the probability simplex of dimension $s$ as: $\mathcal{D}_\mathcal{O} = \{w \in \mathbb{R}^s : \sum_{i=1}^s w_i = 1, w_i \geq 0\}$, which represents the set of all distributions over $\mathcal{O}$.

Examples of annotation $O$: (i) In the *noisy problem*, the true label is replaced (due to mislabeling or corruption) by another label with certain probabilities. Therefore, $\mathcal{O} = \mathcal{Y}$. (ii) In the *superset annotation* problem, the learner observes $o$ which is a subset of $\mathcal{Y}$ (hopefully but not necessarily $o$ contains the true label $y$). In this case, $\mathcal{O} = 2^\mathcal{Y}$, the power set of $\mathcal{Y}$.

In our framework, the learner predicts $O$ using the graphical model shown in fig. 1. The conditional distribution of $O$ given $X = x$ and $Y = y$ can be identified by a mapping from $x$ to a *transition matrix* $T_0(x) := [\mathbb{P}(O = o_j | X = x, Y = y_i)]_{ij}$. A key point of this paper is that in general we do *not* assume that the learner (or learning algorithm) has full information of $T_0(x)$. Instead, we define a *transition hypothesis* to be a candidate transition $T(x)$ (also denoted as $T$ for convenience) that maps the instance $x$ to a stochastic matrix of size $c \times s$. For a fixed $x$, the $i^{\text{th}}$ row of a transition $T(x)$ represents a distribution over $\mathcal{O}$, and is denoted as $(T(x))_i$. The set of all candidate transition hypotheses is called the *transition class*, denoted as $\mathcal{T}$. We assume $T_0 \in \mathcal{T}$. When it is needed to distinguish transition hypothesis from classifiers in $\mathcal{H}$, we will call the latter one a *base hypothesis*. With a transition hypothesis $T$, a base hypothesis $\hat{y} = h(x)$ naturally induces a probability distribution $(T(x))_{h(x)}$. We call it *induced hypothesis*, denoted as $T \circ h$.

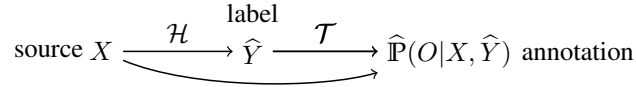

Figure 1: Supervision Model. The learner predicts the label $\widehat{Y}$ of $X$ via $\mathcal{H}$. To supervise using the observation of $(X, O)$, the learner uses $\widehat{Y}$ to induce a probabilistic prediction over $\mathcal{O}$ via $\mathcal{T}$.

One may penalize $T \circ h$ by evaluating its prediction of $O$ on the dataset. More precisely, in our framework, the learner will be penalized by provided with an *annotation loss* $\ell_\mathcal{O}(\hat{y}, T, (x, o))$ : $\mathcal{Y} \times \mathcal{T} \times \mathcal{X} \times \mathcal{O} \mapsto \mathbb{R}$. A natural example is the cross-entropy loss, which approximates the conditional probability of $O$:

$$\ell_\mathcal{O}(h(x), T, (x, o)) := -\log \mathbb{P}(o | x, h(x), T) \tag{1}$$

The *annotation risk* is defined as $R_\mathcal{O}(T \circ h) := \mathbb{E}_{x,o}[\ell_\mathcal{O}(h(x), T, (x, o))]$. A *training set* $S = \{(x^{(i)}, o^{(i)})\}_{i=1}^m$ contains independent samples of $X$ and $O$. The *empirical annotation risk* associated with the training sample $S$ is then defined as $\widehat{R}_\mathcal{O}(h \circ T | S) := \frac{1}{m} \sum_{i=1}^m \ell_\mathcal{O}(h(x^{(i)}), T, (x^{(i)}, o^{(i)}))$.

In summary, the learner's input includes: the spaces $\mathcal{X}, \mathcal{Y}, \mathcal{O}$, the hypothesis class $\mathcal{H}$ and transition class $\mathcal{T}$, the training set $S = \{x^{(i)}, o^{(i)}\}_{i=1}^m$, and the loss functions $\ell, \ell_\mathcal{O}$.

A hypothesis class $\mathcal{H}$ is said to be $(\mathcal{T})$-*learnable* if there is a learning algorithm $\mathcal{A} : \cup_{m=1}^\infty (\mathcal{X} \times \mathcal{O})^m \mapsto \mathcal{H}$ such that: for any distribution $D_X$ over $\mathcal{X}$ and $T_0 \in \mathcal{T}$, when running $\mathcal{A}$ on datasets $S^{(m)}$ of $m$ independent samples of $(X, O)$, we have $R(\mathcal{A}(S^{(m)}))$ converges to 0 in probability as $m \to \infty$. In particular, we define the *Empirical Risk Minimizer* to be a mapping $\text{ERM} : \cup_{m=1}^\infty (\mathcal{X} \times \mathcal{O})^m \mapsto \mathcal{H}$ such that $\text{ERM}(S) \in \operatorname{argmin}_{h \in \mathcal{H}, T \in \mathcal{T}} \widehat{R}_\mathcal{O}(h \circ T | S)$, where the $\operatorname{argmin}$ operator only returns the base hypothesis (although the empirical risk is minimized over both base and transition hypotheses).

# 4 General Learnability Conditions

In this section, we present theorem 4.2 that decomposes the learnability of a general indirect supervision problem into three aspects: *complexity*, *consistency* and *identifiability*. After that, we provide proposition 4.3 to help verifying the complexity condition. The other two conditions will be further studied in the next section.

We assume $\ell_{\mathcal{O}}$ takes value in an interval $[0, b]$ for some constant $b > 0$. To characterize the learnability, a key step is to describe the complexity of the function class

$$\ell_{\mathcal{O}} \circ \mathcal{T} \circ \mathcal{H} \stackrel{\text{def}}{=\!=} \{(x, o) \mapsto \ell_{\mathcal{O}}(T, (x, h(x), o)) : h \in \mathcal{H}, T \in \mathcal{T}\}$$

To do so, we use the following generalized version of VC-dimension proposed in [3]. It will be shown in proposition 4.3 that it enables us to bound the Rademacher complexity [4] of $\ell_{\mathcal{O}} \circ \mathcal{T} \circ \mathcal{H}$ (which provides the flexibility to study arbitrary loss function) via the standard VC dimension or Natarajan dimension [25] (also see Chapter 29 of [7] for an introduction) of $\mathcal{H}$ (which is in general easier to compute than the Rademacher complexity).

**Definition 4.1.** We adopt the following definitions from [3]:

1. (shattering and VC-class) A class $\mathcal{C}$ of subsets of a set $\mathcal{Z}$ is said to *shatter* a finite subset $Z \subseteq \mathcal{Z}$ if

$$\{C \cap Z : C \in \mathcal{C}\} = 2^Z$$

   Moreover, $\mathcal{C}$ is called a *VC-class* with dimension no larger than $k$ if there exists an integer $k$ such that $\mathcal{C}$ cannot shatter any subset of $\mathcal{Z}$ with more than $k$ elements.

2. (weak VC-major) The function class $\ell_{\mathcal{O}} \circ \mathcal{T} \circ \mathcal{H}$ is said to be *weak VC-major* with dimension $d$ if $d$ is the smallest integer such that for all $u \in \mathbb{R}$, the set family

$$\mathcal{C}_u \stackrel{\text{def}}{=\!=} \{\{(x, o) : \ell_{\mathcal{O}}(h(x), T, (x, o)) > u\} : h \in \mathcal{H}, T \in \mathcal{T}\}$$

   is a VC-class of $\mathcal{X} \times \mathcal{O}$ with dimension no larger than $d$.

Now we are able to state the main result of this section:

**Theorem 4.2.** If the following conditions are satisfied

[C1] (Complexity) $\ell_{\mathcal{O}} \circ \mathcal{T} \circ \mathcal{H}$ is weak VC-major with dimension $d < \infty$.

[C2] (Consistency) $h_0 \in \underset{h \in \mathcal{H}, T \in \mathcal{T}}{\operatorname{argmin}} \, {}^2 R_{\mathcal{O}}(T \circ h)$.

[C3] (Identifiability) $\eta \stackrel{\text{def}}{=\!=} \underset{h \in \mathcal{H}, T \in \mathcal{T}: R(h) > 0}{\inf} \dfrac{R_{\mathcal{O}}(T \circ h) - \inf_{T \in \mathcal{T}} R_{\mathcal{O}}(T \circ h_0)}{R(h)} > 0.$

Then, $\mathcal{H}$ is $\mathcal{T}$-learnable. That is, for any $\delta \in (0, 1)$, with probability of at least $1 - \delta$, we have:

$$R(\mathrm{ERM}(S^{(m)})) \leq \frac{2b}{\eta} \left( \sqrt{\frac{2\overline{\Gamma}_m(d)}{m}} + \frac{4\overline{\Gamma}_m(d)}{m} + \sqrt{\frac{2\log(4/\delta)}{m}} \right) \tag{2}$$

where $\overline{\Gamma}_m(d)$ is defined in [3] by

$$\overline{\Gamma}_m(d) \stackrel{\text{def}}{=\!=} \log \left[ 2 \sum_{j=0}^{\min\{d, m\}} \binom{m}{j} \right] = d \log m (1 + o(1)) \text{ as } m \to \infty$$

This implies $R(\mathrm{ERM}(S^{(m)})) \to 0$ in probability as $m \to \infty$.

Bound (2) suggests that the difficulty of the learning can be characterized by (i) the identifiability level $\eta$, which mainly depends on the nature of the indirect supervision signal and the learner's prior information of the transition hypothesis, which will be further studied in the next section. (ii) the weak VC-major $d$ of $\ell_{\mathcal{O}} \circ T \circ \mathcal{H}$, which depends on the modeling choice. We present the following results that bound $d$ by the Natarajan dimension of $\mathcal{H}$ and the weak-VC major dimension of the class:

$$\ell_{\mathcal{O}} \circ \mathcal{T} \stackrel{\text{def}}{=\!=} \{(x, \widehat{y}, o) \mapsto \ell_{\mathcal{O}}(\widehat{y}, T, (x, o)) : T \in \mathcal{T}\}$$

**Proposition 4.3.** Suppose the Natarajan dimension of $\mathcal{H}$ is $d_{\mathcal{H}} < \infty$ and the weak-VC major dimension of $\ell_{\mathcal{O}} \circ \mathcal{T}$ is $d_{\mathcal{T}} < \infty$. Then, the weak-VC major dimension of $\ell_{\mathcal{O}} \circ \mathcal{T} \circ \mathcal{H}$, $d$, can be bounded by:

$$d \leq 2\left((d_{\mathcal{H}} + d_{\mathcal{T}})\log(6(d_{\mathcal{H}} + d_{\mathcal{T}})) + 2d_{\mathcal{H}}\log c\right) \text{ where } c = |\mathcal{Y}|$$

The reason that we do *not* study the complexity of $\mathcal{T}$ separately is that the annotation loss may be independent of $T$ (i.e., $\ell_{\mathcal{O}}(\widehat{y}, T_1, (x, o)) = \ell_{\mathcal{O}}(\widehat{y}, T_2, (x, o))$ for any $T_1, T_2 \in \mathcal{T}$). See proposition 5.5 for an example of such a loss.

To show applications of proposition 4.3, we study the following cases:

**Example 4.4.** In the following cases, we first compute/bound $d_{\mathcal{T}}$, then $d$ can be bounded by $d_{\mathcal{H}}$:

1. When the true transition is known *or* when the annotation loss function only depends on $(\widehat{y}, o)$, we have $d_{\mathcal{T}} = 0$; hence $d \leq 2d_{\mathcal{H}}(\log(6d_{\mathcal{H}}) + 2\log c)$. This is conceptually similar to the Lemma 3.4 in [22], which bounds the VC-dimension of the induced hypothesis class for the noise-free superset problem.

2. When all transition hypotheses in $\mathcal{T}$ are instance-independent *and* the annotation loss only depends on $(T, \widehat{y}, o)$ (e.g., the cross-entropy loss defined in (1)), then $d_{\mathcal{T}}$ can be trivially bounded by $d_{\mathcal{T}} \leq cs = |\mathcal{Y} \times \mathcal{O}|$; hence $d \leq 2((d_{\mathcal{H}} + cs)\log(6(d_{\mathcal{H}} + cs)) + 2d_{\mathcal{H}}\log c)$.

3. Suppose the instance is embedded in a vector space $\mathcal{X} = \mathbb{R}^p$. Consider the problem (Example 5.1.3 in [24]) of binary classification with a uniform noise rate which is modeled as a Logistic regression: $\mathbb{P}(O \neq y | x, y) = S(w^\mathsf{T} x)$ where $S$ is the sigmoid function and $w$ is the parameter. Then the cross-entropy loss becomes: $-\mathbb{1}\{o \neq \widehat{y}\}\log(S(w^\mathsf{T} x)) - \mathbb{1}\{o = \widehat{y}\}\log(1 - S(w^\mathsf{T} x))$. We have $d_{\mathcal{T}} \leq 2p + 2$. See supplementary material for a proof.

# 5 Separation

Throughout this section we assume **[C1]** of Theorem 4.2 holds. We will first propose a concept called *separation*, which provides an intuitive way to understand the learnability and helps to verify **[C2]** and **[C3]**; then we study two ways to ensure separation, and their application in real problems.

## 5.1 Learning by Separation

Without any prior knowledge, the transition class will contain all possible transitions. In this case, learnability cannot be ensured since a wrong label $\widehat{y}$ can also induce a good prediction of $O$ via an incorrect transition hypothesis. Hence, certain kind of prior knowledge is needed to restrict the range of $\mathcal{T}$. To formalize this idea, we first introduce an extension of the KL-divergence.

**Definition 5.1** (KL-divergence between Two Sets of Distributions). Given two sets of distributions $\mathcal{D}_1$ and $\mathcal{D}_2$, we define the *KL-divergence* between them as:

$$\mathrm{KL}(\mathcal{D}_1 \parallel \mathcal{D}_2) \overset{\text{def}}{=\joinrel=} \inf_{D_1 \in \mathcal{D}_1, D_2 \in \mathcal{D}_2} \mathrm{KL}(D_1 \parallel D_2)$$

Now we are able to state the main result of this section:

**Theorem 5.2** (Separation). For all $x \in \mathcal{X}$, we denote the *induced distribution families* by label $y_i$ as $\mathcal{D}_i(x) \overset{\text{def}}{=\joinrel=} \{(T(x))_i : T \in \mathcal{T}\} \subseteq \mathcal{D}_{\mathcal{O}}$ (recall that $(T(x))_i$ is the $i^{\text{th}}$ row of $T(x)$), and the set of all possible predictions of the label as $\mathcal{H}(x) \overset{\text{def}}{=\joinrel=} \{h(x) : h \in \mathcal{H}\} \subseteq \mathcal{Y}$. Suppose

$$\gamma \overset{\text{def}}{=\joinrel=} \inf_{(x,i,j):p(x,y_i)>0, j\neq i, y_j \in \mathcal{H}(x)} \mathrm{KL}(\mathcal{D}_i(x) \parallel \mathcal{D}_j(x)) > 0 \tag{3}$$

Then $\mathcal{H}$ is learnable from the observations of $(X, O)$ with $\eta \geq \gamma > 0$ via the ERM of cross-entropy loss (1). We call $\gamma$ the *separation degree*.

Moreover, if (3) is not satisfied, then there exists a sequence of transitions $\{T^{(k)}\}_k$ $(T^{(k)} \in \mathcal{T})$ and distributions $\{D_X^{(k)}\}_k$ over $\mathcal{X}$ such that $\lim_k \eta^{(k)} = 0$ , where $\eta^{(k)}$ is defined the same as $\eta$ in **[C3]**, with the expectation (in the definition of the risk functions) being taken according to $T^{(k)}$ and $D_X^{(k)}$.

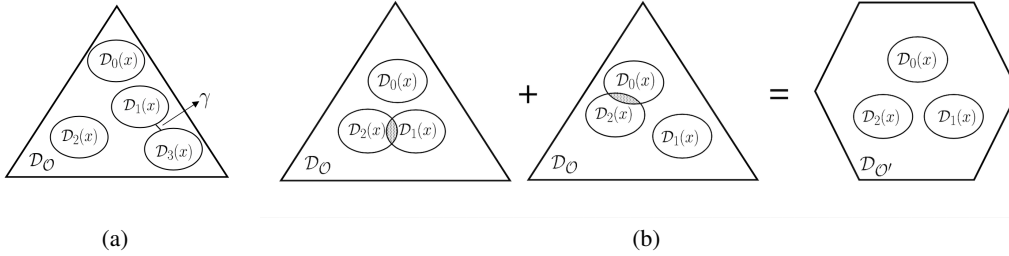

(a)                                                  (b)

Figure 2: **(a)** Illustration of *separation*. A (predicted) label $y_i$ will induce a distribution family over $\mathcal{O}$ called $\mathcal{D}_i(x)$. Different families are separated by a minimal "distance" $\gamma$. **(b)** Illustration of *joint supervision*. By adding new supervision signals, separation of particular pairs of labels is preserved.

Theorem 5.2 is important in two ways: (i) It provides a way to characterize the prior knowledge of the learner about the transition using the KL-divergence and reveals its connection with the identifiability of labels. (ii) The "moreove" result shows if separation is not satisfied, then the induced distribution of $O$ by different labels can be arbitrarily close, and hence the learning of $Y$ from $O$ can be arbitrarily difficult. An illustration of separation is shown in fig. 2 (a). Yet, a drawback of the cross-entropy loss used in theorem 5.2 is that it can be unbounded when there is a zero element in the transition matrix. This problem will be partly solved in proposition 5.5 by introducing a different annotation loss.

As the simplest application, we introduce the case where the transition is fully known to the learner. In this case, the induced distribution families $\mathcal{D}_i(x)$ reduce to some points in the probability simplex.

**Example 5.3** (Full Information of the Transition). Suppose the transition $T(x)$ is known to the learner, i.e., $\mathcal{T} = \{T_0\}$, by Theorem 5.2, we know $\mathcal{H}$ is learnable if

$$\inf_{(x,i,j):p(x)>0,\ i\neq j, y_j \in \mathcal{H}(x)} \mathrm{KL}((T_0(x))_i \parallel (T_0(x))_j) > 0$$

Notice that this is a weaker assumption than the invertibility assumption of $T_0(x)$, which is used in [36] (called *reconstructible corruption*). This is possible because we assume a deterministic rule for $X \to Y$ but a randomized process for $X, Y \to O$, hence the latter one could contain more information and is capable of encoding the deterministic rule even when $\mathcal{O}$ is smaller than $\mathcal{Y}$. For a concrete example, consider the following constant transition matrix:

$$T = \begin{bmatrix} 0.1 & 0.9 \\ 0.5 & 0.5 \\ 0.9 & 0.1 \end{bmatrix}$$

In this case, $|\mathcal{Y}| = 3 > |\mathcal{O}| = 2$ and therefore $T$ is not right-invertible. However, since the distribution of $O$ induced by each labels $Y$ is known to learner, and this distribution could be estimated from the observation of $O$, the learner is able to recover the true label from the indirect supervision.

## 5.2 Separation by Total Variation

In this subsection, we introduce a way to guarantee separation by controlling the KL-divergence using total variation distance, which is done via the well-known Pinsker's inequality [35]:

**Lemma 5.4** (Pinsker's inequality, proposed in [28], see [35] for an introduction). If $P$ and $Q$ are two probability distributions on the same measurable space $(\Omega, \mathcal{F})$, then

$$\|P - Q\|_{\mathrm{TV}} \leq \sqrt{\mathrm{KL}(P \parallel Q)/2}$$

where $\|\cdot\|_{\mathrm{TV}}$ is the total variance distance: $\|P - Q\|_{\mathrm{TV}} \overset{\text{def}}{=\!=} \sup_{A \in \mathcal{F}} \|P(A) - Q(A)\|$. Moreover, if $\Omega$ is countable (in our case, $\Omega = \mathcal{O}$ is finite and hence, then the total variance distance is equivalent to the $L^1$-distance in the sense that $\|P - Q\|_{\mathrm{TV}} = \frac{1}{2}\|P - Q\|_1$).

This lemma suggests we can ensure separation by controlling the $L^1$-distance. To show a concrete example, we introduce the *concentration* condition. The intuition behind it is that the information of different labels in $\mathcal{Y}$ is concentrated in relatively different sets of annotations. Formally:

**Proposition 5.5** (Concentration). A sufficient condition for (3) is that for every $1 \leq i \leq c$, there exists a set $S_i \subset \mathcal{O}$ (we call them *concentration sets*) such that

$$\gamma_C \overset{\text{def}}{=\joinrel=} \inf_{(i,j,x,T):T\in\mathcal{T},p(x)>0,j\neq i} \mathbb{P}_T(O \in S_i|x,y_i) - \mathbb{P}_T(O \in S_j|x,y_i) > 0 \tag{4}$$

where $\mathbb{P}_T(\cdot)$ is the conditional probability defined by transition $T$. Under this condition, we can relate identifiability and separation degree by $\eta \geq \gamma \geq 2\gamma_C^2$. Since a condition imposed on all $T \in \mathcal{T}$ can be regarded as an assumption imposed on the true transition $T_0$, condition (4) can be rewritten as:

$$\gamma_C = \inf_{(i,j,x):p(x,y_i)>0,i\neq j} \mathbb{P}(O \in S_i|x,y_i) - \mathbb{P}(O \in S_j|x,y_i) > 0 \tag{5}$$

Also, in this case, one can ensure learnability by the ERM which minimizes the following transition-independent annotation loss

$$\ell_{\mathcal{O}}(h(x),T,(x,o)) \overset{\text{def}}{=\joinrel=} \mathbb{1}\{o \notin S_{h(x)}\} \tag{6}$$

For this annotation loss, we can bound the identifiability level by $\eta \geq \gamma_C$.

**Example 5.6** (Superset with Noise). For superset with noise problem where $O$ is a random subset of $\mathcal{Y}$ (i.e., $\mathcal{O} = 2^{\mathcal{Y}}$), let $S_i = \{o : y_i \in o\} \subset \mathcal{O}$, the conditional (5) becomes

$$\gamma_C = \inf_{p(x,y_i)>0,i\neq j} \mathbb{P}(y_i \in O|x,y_i) - \mathbb{P}(y_j \in O|x,y_i) > 0 \tag{7}$$

This generalizes the *small ambiguity degree condition* proposed in [16, 22], which assumes $\mathbb{P}(y_i \in O|x,y_i) = 1$ (i.e., the gold label always lies in the superset). [16, 22] also proposes a *superset loss*, which is the special case of (6). We extend the discussion to allow the presence of noise.

The following example can be regarded as a special case of Example 5.6.

**Example 5.7** (Label Noise). For noisy problem where $\mathcal{O} = \mathcal{Y}$, let $S_i = \{y_i\}$, condition (4) becomes

$$\gamma_C = \inf_{p(x,y_i)>0,i\neq j} \mathbb{P}(O = y_i|x,y_i) - \mathbb{P}(O = y_j|x,y_i) > 0 \tag{8}$$

This generalizes the Massart noise condition [23] of binary classification, which assumes the noise rate is lower bounded by $1/2$ minus a constant. We extend the discussion to multiclass case.

Also, notice that (6) is simply the zero-one loss for $O$, which means learnability can still be guaranteed if one ignores the noisy process and learns $O$ as clean label. This partly explains the empirical study in [31], which tests the robustness of neural networks (without additional denoising process) to noise in annotations. [31] proposes a parameter called $\delta$-degree which is similar to $\gamma_C$ and observes that the performance of the network decreases as $\delta$ decreases, as our learning bound (2) suggests.

We can further generalize proposition 5.5 by encoding *functional* prior information of the transition:

**Proposition 5.8** (Evidence). A sufficient condition for (3) is that there exists Lipschitz (with respect to the $L^1$-norm of vectors) functions $\Phi_{ij} : \mathbb{R}^c \to \mathbb{R}, 1 \leq i \neq j \leq c$ (we call them *evidence*) with Lipschitz constants $L_{ij}$ such that

$$\gamma_{ij} \overset{\text{def}}{=\joinrel=} \inf_{p(x,y_i)>0,y_j\in\mathcal{H}(x),D_i\in\mathcal{D}_i(x),D_j\in\mathcal{D}_j(x)} \Phi_{ij}(D_i) - \Phi_{ij}(D_j) > 0 \tag{9}$$

In this case, the separation degree can be bounded by $\gamma \geq 1/2 \min_{i\neq j} (\gamma_{ij}/L_{ij})^2$.

In particular, the dot product with a fixed vector $\Phi_u(t) = \langle u, t \rangle$ ($\langle \cdot, \cdot \rangle$ represents the dot product) is Lipschitz with Lipschitz constant $L_u \leq \|u\|_\infty$. As an example, given sets $S_i \subset \mathcal{O}$ ($1 \leq i \leq c$), letting $\Phi_{ij}(\cdot) = \langle \sum_{k:o_k \in S_i} \hat{e}_k - \sum_{k:o_k \in S_j} \hat{e}_k, \cdot \rangle$ recovers proposition 5.5, where $\hat{e}_k$ is the $k^{\text{th}}$ standard unit basis vector of $\mathbb{R}^c$. Another example will be given in Example 5.13.

## 5.3 Separation by Joint Supervision

When a weak supervision signal cannot ensure learnability individually, it needs to be used with other forms of annotations together to supervise the learning. Our goal in this subsection is to provide a way to describe the effect of using multiple sources of annotations jointly. We will show that joint supervision can improve (Example 5.13), preserve (Proposition 5.10) or even damage (Remark 5.11) the separation.

First, we formulate the joint supervision problem. For simplicity, we only consider the case that we have two sources of annotations $O_1, O_2$, and the general case can be discussed in a similar way. For each $O_k, k \in \{1,2\}$, denote its annotation space as $\mathcal{O}_k$, its transition as $T_k(x)$ and its transition classes as $\mathcal{T}_k$. We focus on the scenario that for each instance $x$, there is only *one* type of annotation. Then the joint annotation space is $\mathcal{O} = \mathcal{O}_1 \cup \mathcal{O}_2$. We model the *annotation type* $\mathbb{1}\{O = O_k\}$ as a random variable that is independent with $X$ and all the $O_k$, and the probability $\mathbb{P}(O = O_1) = \lambda$ is known to the learner. Then the *joint annotation* is defined as: $O = \mathbb{1}\{O = O_1\}O_1 + \mathbb{1}\{O = O_2\}O_2$.

Next, we quantify the supervision power of an annotation if separation is not guaranteed via a local version of the separation (degree):

**Definition 5.9** (Pairwise Separation). Define the *separation degree of $y_i$ to $y_j$* as

$$\gamma_{i \to j} \overset{\text{def}}{=\!=} \inf_{x:p(x,y_i)>0, y_j \in \mathcal{H}(x)} \mathrm{KL}(\mathcal{D}_i(x) \| \mathcal{D}_j(x)) \tag{10}$$

We say the labels $y_i$ is *separated from* a $y_j$ if $\gamma_{i \to j} > 0$. The separation degree $\gamma = \min_{i,j} \gamma_{i \to j}$.

This definition gives a probabilistic formulation of the intuition that a (weak) supervision signal can help *distinguish* certain pairs of labels. For example, a noisy annotation for multiclass classification may break the condition (8) due to a large noise rate for certain labels, but it can still provide information to separate other labels if (8) is satisfied for any other pairs of $(i, j)$.

When there are no additional constraints on the joint transition, one can construct the joint transition simply by combining the candidate transitions in $\mathcal{T}_1, \mathcal{T}_2$. For example, the induced distribution family by $y_i$ of joint supervision can be naturally constructed by

$$\mathcal{D}_i(x) = \{\lambda D_1 + (1 - \lambda)D_2 : D_1 \in \mathcal{D}_{i1}(x), D_2 \in \mathcal{D}_{i2}(x)\} \tag{11}$$

where $\mathcal{D}_{i1}$ and $\mathcal{D}_{i2}$ are the induced distribution family by $y_i$ of $O_1$ and $O_2$. In this case, we present the following result to characterize the learnability under joint supervision $O$:

**Proposition 5.10** (No Free Separation). Suppose the separation degrees of $y_i$ to $y_j$ of $O_1$ and $O_2$ are $\gamma_{i \to j1}$ and $\gamma_{i \to j2}$ respectively. Then, if the joint transition class is constructed as (11), then the separation degrees of $y_i$ to $y_j$ for the joint supervision satisfies:

$$\gamma_{i \to j} \leq \lambda \gamma_{i \to j1} + (1 - \lambda)\gamma_{i \to j2}$$

Also, if $\mathcal{O}_1 \cap \mathcal{O}_2 = \varnothing$, then the two equality holds. As a consequence, a necessary condition of that $y_i$ is separated from $y_j$ by the joint signal $O$ is that $y_i$ must be separated from $y_j$ by one of $O_1, O_2$.

**Remark 5.11** (Defining $\mathcal{O}_1 \cap \mathcal{O}_2$). The condition $\mathcal{O}_1 \cap \mathcal{O}_2 = \varnothing$ means that the learner *distinguishes* different annotations. For example, in a crowdsourcing setting, we have two annotators and each provides a noisy annotation, then $\mathcal{O}_1, \mathcal{O}_2 = \mathcal{Y}$. But as long as the learner distinguishes the annotations of the two annotators, we can nevertheless write $\mathcal{O}_1 \cap \mathcal{O}_2 = \varnothing$. Without this condition, even if both $\gamma_{i \to j1}, \gamma_{i \to j2} > 0$, we can still have $\gamma_{i \to j} = 0$. See the supplementary material for an example. This idea has also been explored in the empirical study of [18], which observes that in a crowdsourcing setting, the model performance improves if annotator identifiers are input as features. However, one should note that the tradeoff is the model complexity: distinguishing different annotations will in general require more parameters to model the joint transition.

**Remark 5.12.** Proposition 5.10 shows that without constraints, the joint supervision does not create new separation, however, it can preserve the separation between labels by the original supervision signals. So in this view, the weak supervision signal can be regarded as a "building block" for the (global) separation (3) by contributing pairwise separation (10). An illustration is shown in fig. 2 (b).

If there does exist constraints about the two transition classes, Proposition 5.10 no longer holds and joint supervision may create new separation. To illustrate, consider the following artificial example:

**Example 5.13** (Learning from Difference). Given a binary classification problem where $\mathcal{Y} = \{\pm 1\}$. Suppose we have two annotators $O_1$ and $O_2$ and each provides a noisy annotation with an unknown, uniform, instance-independent noise, i.e., $\eta_1 \overset{\text{def}}{=\!=} \mathbb{P}(O_1 \neq y|x, y = -1) = \mathbb{P}(O_1 \neq y|x, y = +1)$, $\eta_2 \overset{\text{def}}{=\!=} \mathbb{P}(O_2 \neq y|x, y = -1) = \mathbb{P}(O_2 \neq y|x, y = +1)$, where $\eta_1, \eta_2$ are constants independent of $x$. Then, the joint transition is modeled as:

$$T = \begin{bmatrix} \lambda(1 - \eta_1) & \lambda\eta_1 & (1 - \lambda)(1 - \eta_2) & (1 - \lambda)\eta_2 \\ \lambda\eta_1 & \lambda(1 - \eta_1) & (1 - \lambda)\eta_2 & (1 - \lambda)(1 - \eta_2) \end{bmatrix} = \begin{bmatrix} D_1 \\ D_2 \end{bmatrix}$$

Now, suppose it is known that the first annotator provides a better quality of annotation, i.e., there is a $\gamma \in \mathbb{R}$ (known to the learner) such that $\eta_1 - \eta_2 \leq \gamma < 0$. To apply proposition 5.8, define the evidence $\Phi(\cdot) = \langle \hat{e}_1/\lambda - \hat{e}_3/(1-\lambda), \cdot \rangle$, then for any $D_1, D_2$, we have $\Phi(D_1) = \eta_2 - \eta_1 \geq \gamma$ and $\Phi(D_2) = \eta_1 - \eta_2 \leq -\gamma$. So by proposition 5.8, the original classification hypothesis is learnable. Notice that without joint supervision, separation is not guaranteed since we have no restriction on $\eta_1$ or $\eta_2$ individually.

This example shows the necessity to model possible constraints between different supervision sources, which help to reduce the size of the joint transition class and may improve the separation degree.

## 6 Conclusion and Future Work

In this paper, we provide a unified framework for analyzing the learnability of multiclass classification with indirect supervision. Our theory builds upon two key components: (i) The construction of the induced hypothesis class and its complexity analysis, which allows us to indirectly supervise the learning by minimizing the annotation risk. (ii) A formal description of the learner's prior knowledge about the transition and its encoding in the learning condition, which allows us to bound the classification error by the annotation risk.

The notion of separation depends on the annotation loss being used. The KL-divergence may be replaced by other statistical distances, as long as the distance can induce a loss function. However, the idea behind separation is invariant: the prior knowledge needs to be strong enough to distinguish different labels via the observables. Moreover, theorem 5.2 shows that separation is a sufficient and almost necessary condition, and the later examples show separation is also practically useful and can easily produce learnability conditions. Therefore, we believe the the concepts introduced are general, and that our analysis tools can be applied in many other supervision scenarios.

One limitation of our work is that the definition of learnability requires us to handle every possible $\mathcal{D}_X$, and the consequence is that we need to ensure separation at every $x \in \mathcal{X}$. In future work, we may try to relax the learnability conditions by encoding prior knowledge of $\mathcal{D}_X$, which can be obtained from unlabeled datasets. Another direction to explore is to extend the discussion to the agnostic case as well as the case where $T_0 \notin \mathcal{T}$. Also, in a directly supervised learning setting, classical realizable PAC learning could achieve a convergence rate of $O(\log(1/\delta)/m)$, which is better than the rate of $O(\sqrt{\log(1/\delta)/m})$ we derived for a general indirect supervision problem. It is worth exploring whether and how our bounds could be improved for more kinds of supervision signals (other than the gold label).

## Broader Impact

Our work mostly focuses on theoretical aspects of learning, however, it provides better understanding and thus can suggest new machine learning scenarios and algorithms for learning from indirect observations; this addresses a key challenge to machine learning today, and will help machine learning researchers to reduce the cost of and need for labeled data. Our theory may have positive and negative impact on the privacy protection of sensitive data. On one hand, the theory suggests that one can alter the forms of data (via a probabilistic transition) to ensure privacy while keeping its usefulness (learnability). On the other hand, it might be possible for an attacker to recover sensitive information about the data indirectly through a related dataset.

## Acknowledgments and Disclosure of Funding

This work was supported by the Army Research Office under Grant Number W911NF-20-1-0080 and by contract FA8750-19-2-0201 with the US Defense Advanced Research Projects Agency (DARPA). The views expressed are those of the authors and do not reflect the official policy or position of the Department of Defense or the U.S. Government.

## Footnotes

[2]This argmin operator only returns the base hypothesis.

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
