[Supplementary Material]

# Supplementary Material for Submission ID 8000: Learnability with Indirect Supervision Signals

**Kaifu Wang**
University of Pennsylvania
kaifu@sas.upenn.edu

**Qiang Ning**[*]
Amazon
qning@amazon.com

**Dan Roth**
University of Pennsylvania
danroth@seas.upenn.edu

## 1 Proofs

### 1.1 Proof of Theorem 4.2

**Theorem.** If the following conditions are satisfied

[C1] (Complexity) $\ell_{\mathcal{O}} \circ \mathcal{T} \circ \mathcal{H}$ is weak VC-major with dimension $d < \infty$.

[C2] (Consistency) $h_0 \in \underset{h \in \mathcal{H}, T \in \mathcal{T}}{\operatorname{argmin}} {}^2 R_{\mathcal{O}}(T \circ h)$.

[C3] (Identifiability) $\eta \overset{\text{def}}{=} \underset{h \in \mathcal{H}, T \in \mathcal{T}: R(h) > 0}{\inf} \dfrac{R_{\mathcal{O}}(T \circ h) - \inf_{T \in \mathcal{T}} R_{\mathcal{O}}(T \circ h_0)}{R(h)} > 0$.

Then, $\mathcal{H}$ is $\mathcal{T}$-learnable. That is, for any $\delta \in (0, 1)$, with probability of at least $1 - \delta$, we have:

$$R(\text{ERM}(S^{(m)})) \leq \frac{2b}{\eta} \left( \sqrt{\frac{2\overline{\Gamma}_m(d)}{m}} + \frac{4\overline{\Gamma}_m(d)}{m} + \sqrt{\frac{2\log(4/\delta)}{m}} \right)$$

where $\overline{\Gamma}_m(d)$ is defined in [1] as:

$$\overline{\Gamma}_m(d) \overset{\text{def}}{=} \log \left[ 2 \sum_{j=0}^{d \wedge m} \binom{m}{j} \right] = d \log m (1 + o(1)) \text{ as } m \to \infty \tag{1}$$

where $d \wedge m = \min\{d, m\}$. This implies $R(\text{ERM}(S^{(m)})) \to 0$ in probability as $m \to \infty$.

We need several intermediate results to prove this. First, we introduce the definition of the averaged Rademacher complexity.

**Definition 1.1** (Averaged Rademacher Complexity [2]). The averaged Rademacher complexity [2] of $\ell_{\mathcal{O}} \circ \mathcal{T} \circ \mathcal{H}$ with respect to $m$ samples is defined as

$$\Re_m(\ell_{\mathcal{O}} \circ \mathcal{T} \circ \mathcal{H}) \overset{\text{def}}{=} \mathbb{E}_{\epsilon, x, o} \left[ \frac{1}{m} \sup_{h \in \mathcal{H}, h \in \mathcal{T}} \left| \sum_{i=1}^{m} \epsilon^{(i)} \ell_{\mathcal{O}}(h(x^{(i)}), T, (x^{(i)}, o^{(i)})) \right| \right] \tag{2}$$

where $\epsilon_i \overset{\text{iid}}{\sim} \text{Uniform}\{-1, +1\}$ are the so-called Rademacher random variables and the expectation is taken over $m$ i.i.d. samples of $\epsilon, x, o$ .

The first lemma bounds the empirical risk via the averaged Rademacher complexity.

---

[*]Work done while at the Allen Institute for AI and at the University of Illinois at Urbana-Champaign.

[2]This argmin operator only returns the base hypothesis.

**Lemma 1** (Adapted from the proof of Theorem 26.5 in [3]). *In this lemma and its proof, for convenience, we let the ERM algorithm return the induced hypothesis in $\mathcal{H} \circ \mathcal{T}$ (rather than the base hypothesis only).*

Given any $\delta \in (0, 1)$, with probability of at least $1 - \delta$, we have

$$R_{\mathcal{O}}(\mathrm{ERM}(S^{(m)})) - \inf_{h, T} R_{\mathcal{O}}(T \circ h) \leq 2\Re_m(\ell_{\mathcal{O}} \circ \mathcal{T} \circ \mathcal{H}) + 2b\sqrt{\frac{2\log(4/\delta)}{m}}$$

*Proof.* Let $T^\star \circ h^\star$ be any induced hypothesis in $\mathcal{T} \times \mathcal{H}$. Given dataset $S^{(m)}$, we have,

$$R_{\mathcal{O}}(\mathrm{ERM}(S^{(m)})) - R_{\mathcal{O}}(T^\star \circ h^\star)$$
$$= R_{\mathcal{O}}(\mathrm{ERM}(S^{(m)})) - \widehat{R}_{\mathcal{O}}(\mathrm{ERM}(S^{(m)})) + \underbrace{\widehat{R}_{\mathcal{O}}(\mathrm{ERM}(S^{(m)})) - \widehat{R}_{\mathcal{O}}(T^\star \circ h^\star)}_{\leq 0}$$
$$+ \widehat{R}_{\mathcal{O}}(T^\star \circ h^\star) - R_{\mathcal{O}}(T^\star \circ h^\star)$$
$$\leq R_{\mathcal{O}}(\mathrm{ERM}(S^{(m)})) - \widehat{R}_{\mathcal{O}}(\mathrm{ERM}(S^{(m)})) + \widehat{R}_{\mathcal{O}}(T^\star \circ h^\star) - R_{\mathcal{O}}(T^\star \circ h^\star)$$

By Theorem 26.5 (i) of [3], we have that with probability of at least $1 - \delta/2$,

$$R_{\mathcal{O}}(\mathrm{ERM}(S^{(m)})) - \widehat{R}_{\mathcal{O}}(\mathrm{ERM}(S^{(m)})) \leq 2\Re'_m(\ell_{\mathcal{O}} \circ \mathcal{T} \circ \mathcal{H}) + b\sqrt{\frac{2\log(4/\delta)}{m}}$$

where $\Re'_m(\ell_{\mathcal{O}} \circ \mathcal{T} \circ \mathcal{H})$ is defined slightly differently in [3] as:

$$\Re'_m(\ell_{\mathcal{O}} \circ \mathcal{T} \circ \mathcal{H}) \stackrel{\text{def}}{=\!=} \mathbb{E}_{\epsilon, x, o}\left[\frac{1}{m}\sup_{h \in \mathcal{H}, h \in \mathcal{T}} \sum_{i=1}^{m} \epsilon^{(i)} \ell_{\mathcal{O}}(h(x^{(i)}), T, (x^{(i)}, o^{(i)}))\right] \qquad (3)$$

It can be seen that $\Re'_m(\ell_{\mathcal{O}} \circ \mathcal{T} \circ \mathcal{H}) \leq \Re_m(\ell_{\mathcal{O}} \circ \mathcal{T} \circ \mathcal{H})$ since the two quantities only differ by the absolute value. Hence

$$R_{\mathcal{O}}(\mathrm{ERM}(S^{(m)})) - \widehat{R}_{\mathcal{O}}(\mathrm{ERM}(S^{(m)})) \leq 2\Re_m(\ell_{\mathcal{O}} \circ \mathcal{T} \circ \mathcal{H}) + b\sqrt{\frac{2\log(4/\delta)}{m}}$$

Also, by Hoeffding's inequality, we have that with probability of at least $1 - \delta/2$,

$$\widehat{R}_{\mathcal{O}}(T^\star \circ h^\star) - R_{\mathcal{O}}(T^\star \circ h^\star) \leq b\sqrt{\frac{\log(4/\delta)}{2m}}$$

Combining the inequalities, we have that with probability of at least $1 - \delta$,

$$R_{\mathcal{O}}(\mathrm{ERM}(S^{(m)})) - R_{\mathcal{O}}(T^\star \circ h^\star) \leq 2\Re_m(\ell_{\mathcal{O}} \circ \mathcal{T} \circ \mathcal{H}) + 2b\sqrt{\frac{2\log(4/\delta)}{m}}$$

Since the above inequality holds for any $T^\star \circ h^\star \in \mathcal{T} \times \mathcal{H}$, taking infimum for all $T^\star \circ h^\star$ gives the desired result. $\qquad \square$

The second lemma bounds the averaged Rademacher complexity via the weak VC-major, which is provided in [1].

**Lemma 2** (Adapted from the Theorem 2.1 in [1]). *Suppose the weak VC-major dimension of $\ell_{\mathcal{O}} \circ \mathcal{T} \circ \mathcal{H}$ is $d$. then,*

$$m\Re_m(\ell_{\mathcal{O}} \circ \mathcal{T} \circ \mathcal{H}) \leq \sigma \log\left(\frac{eb}{\sigma}\right)\sqrt{2m\overline{\Gamma}_m(d)} + 4b\overline{\Gamma}_n(d) \qquad (4)$$

where e is the base of the natural logarithm and

$$\sigma \stackrel{\text{def}}{=\!=} \sup_{h \in \mathcal{H}, T \in \mathcal{T}} \sqrt{\mathbb{E}_{x, o}[\ell_{\mathcal{O}}^2(T, (x, h(x), o))]} \in (0, b] \qquad (5)$$

*Proof.* The proof of the Theorem 2.1 in [1] is long and is presented in the section 3 of [1]. Here we only point out how to use Theorem 2.1 of [1] (equation (2.8) of the paper) to derive our lemma.

First, the Theorem 2.1 of [1] bounds an empirical process (denoted as $\mathbb{E}[Z(\mathcal{F})]$ in the paper, where $\mathcal{F}$ is a function class and here we let $\mathcal{F} = \ell_{\mathcal{O}} \circ \mathcal{T} \circ \mathcal{H}$) rather than the averaged Rademacher complexity (denoted as $\mathbb{E}[\overline{Z}(\mathcal{F})]$ in the paper). However, the proof of Theorem 2.1 of [1] aims to bound the averaged Rademacher complexity $\mathbb{E}[\overline{Z}(\mathcal{F})]$ and then uses the relation $\mathbb{E}[Z(\mathcal{F})] \leq 2\mathbb{E}[\overline{Z}(\mathcal{F})]$ (Lemma 2.1 of [1]) to obtain the bound for $\mathbb{E}[Z(\mathcal{F})]$. Therefore, the proof of the Theorem 2.1 in [1] tells:

$$\mathbb{E}[\overline{Z}(\mathcal{F})] = m\Re_m(\ell_{\mathcal{O}} \circ \mathcal{T} \circ \mathcal{H}) \leq \sigma \log\left(\frac{e}{\sigma}\right)\sqrt{2m\overline{\Gamma}_m(d)} + 4\overline{\Gamma}_n(d) \qquad (6)$$

Second, in the Theorem 2.1 of [1], it is assumed that the functions in $\mathcal{F}$ is bounded in the interval $[0,1]$. Hence, we need scale the annotation loss to $\ell_{\mathcal{O}}/b$ in order to use the theorem (i.e., let $f = \ell_{\mathcal{O}}/b$ in the definition of $\mathbb{E}[Z(\mathcal{F})]$, i.e., equation (1.2) of [1]). Also, in this case, the supreme of variance (5) is scaled to $\sigma/b$. So, the inequality (6) is rewritten as:

$$\mathbb{E}[\overline{Z}(\mathcal{F})] = \frac{m}{b}\Re_m(\ell_{\mathcal{O}} \circ \mathcal{T} \circ \mathcal{H}) \leq \frac{\sigma}{b} \log\left(\frac{e}{\sigma/b}\right)\sqrt{2m\overline{\Gamma}_m(d)} + 4\overline{\Gamma}_n(d) \qquad (7)$$

Rearranging the inequality gives the desired result. $\qquad\square$

Now, we are able to give the proof of the original theorem:

*Proof.* By consistency [C2], we have

$$\inf_{h,T} R_{\mathcal{O}}(T \circ h) = \inf_T R_{\mathcal{O}}(T \circ h_0)$$

Therefore, by lemma 1, we have that with probability of at least $1 - \delta$,

$$R_{\mathcal{O}}(\text{ERM}(S^{(m)})) - \inf_T R_{\mathcal{O}}(T \circ h_0) \leq 2\Re_m(\ell_{\mathcal{O}} \circ \mathcal{T} \circ \mathcal{H}) + 2b\sqrt{\frac{2\log(4/\delta)}{m}}$$

By identifiability [C3], we have that with probability of at least $1 - \delta$,

$$R(\text{ERM}(S^{(m)})) \leq \frac{1}{\eta}\left(2\Re(\ell_{\mathcal{O}} \circ \mathcal{T} \circ \mathcal{H}) + 2b\sqrt{\frac{2\log(4/\delta)}{m}}\right) \qquad (8)$$

By [C1] and lemma 2, we bound the Rademacher Complexity by

$$\Re_m(\ell_{\mathcal{O}} \circ \mathcal{T} \circ \mathcal{H}) \leq \sigma \log\left(\frac{eb}{\sigma}\right)\sqrt{\frac{2\overline{\Gamma}_m(d)}{m}} + 4\frac{b}{m}\overline{\Gamma}_n(d)$$

$$\leq b\sqrt{\frac{2\overline{\Gamma}_m(d)}{m}} + 4\frac{b}{m}\overline{\Gamma}_n(d) \qquad (9)$$

Now the result follows by combining (8) and (9). $\qquad\square$

## 1.2 Proof of Proposition 4.3

**Proposition.** Suppose the Natarajan dimension of $\mathcal{H}$ is $d_{\mathcal{H}} < \infty$ and the weak-VC major dimension of $\ell_{\mathcal{O}} \circ \mathcal{T}$ is $d_{\mathcal{T}} < \infty$. Then, the weak-VC major dimension of $\ell_{\mathcal{O}} \circ \mathcal{H}$, $d$, can be bounded:

$$d \leq 2\left((d_{\mathcal{H}} + d_{\mathcal{T}})\log(6(d_{\mathcal{H}} + d_{\mathcal{T}})) + 2d_{\mathcal{H}}\log c\right) \text{ where } c = |\mathcal{Y}|$$

*Proof.* First, we translate weak-VC major to the language of standard VC-dimension [6]: For a fixed $u \in \mathbb{R}$ and every $h \in \mathcal{H}, T \in \mathcal{T}$, we define an binary classifier: $f_{h,T,u}(x,o) = \mathbb{1}\{\ell_{\mathcal{O}}(h(x), T, (x,o)) > u\}$ and denote $\mathcal{F}_u := \{f_{h,T,u} : h \in \mathcal{H}, T \in \mathcal{T}\}$ as the set of such classifiers. Then $\mathcal{C}_u$ *shatters* a set in $\mathcal{X} \times \mathcal{O}$ if and only if $\mathcal{F}_u$ shatters (in VC theory) the same set, so $\ell_{\mathcal{O}} \circ \mathcal{T} \circ \mathcal{H}$ is weak VC-major with dimension $d$ if $d = \max_{u \in \mathbb{R}} \text{VC}(\mathcal{F}_u) < \infty$, where $\text{VC}(\cdot)$ is the VC dimension for hypothesis class of binary classifiers.

Let $M$ be the maximum number of distinct ways to classify $d$ points in $\mathcal{X}$ by $\mathcal{H}$. Then for $d$ points in $\mathcal{X} \times \mathcal{O}$, suppose there are at most $M$ ways to assign multi-class labels $Y$ to each point. By Natarajan's lemma [5] of multiclass classification, we have

$$M \leq d^{d_\mathcal{H}} c^{2d_\mathcal{H}} \tag{10}$$

For each way of assignment, it produces a set of $d$ points in $\mathcal{X} \times \mathcal{Y} \times \mathcal{O}$, and for these $d$ points, by Sauer-Shelah lemma, there are at most

$$\sum_{i=0}^{d_\mathcal{T}} \binom{d}{i} \leq \left( \frac{ed}{d_\mathcal{T}} \right)^{d_\mathcal{T}}$$

ways to classify if $\ell_\mathcal{O}(\widehat{y}, T, (x, o)) > u$ by $\mathcal{T}$, so in total we have

$$2^d \leq M \sum_{i=0}^{d_\mathcal{T}} \binom{d}{i} \leq M \left( \frac{ed}{d_\mathcal{T}} \right)^{d_\mathcal{T}}$$

where e is the base of the natural logarithm. Therefore, $M \geq 2^d \left( d_\mathcal{T}/ed \right)^{d_\mathcal{T}}$. Then, by (10)

$$d^{d_\mathcal{H}} c^{2d_\mathcal{H}} \geq M \geq 2^d \left( \frac{d_\mathcal{T}}{ed} \right)^{d_\mathcal{T}}$$

Taking logarithm in both side, we have

$$d_\mathcal{H} \log d + 2d_\mathcal{H} \log c \geq d \log 2 + d_\mathcal{T} (\log d_\mathcal{T} - \log d - 1)$$

Rearrange the inequality,

$$\begin{aligned}
d \log 2 + d_\mathcal{T} (\log d_\mathcal{T} - 1) &\leq (d_\mathcal{H} + d_\mathcal{T}) \log d + 2d_\mathcal{H} \log c \\
&\leq (d_\mathcal{H} + d_\mathcal{T}) \left( \frac{d}{6(d_\mathcal{H} + d_\mathcal{T})} + \log(6(d_\mathcal{H} + d_\mathcal{T})) - 1 \right) + 2d_\mathcal{H} \log c \\
&= d/6 + (d_\mathcal{H} + d_\mathcal{T}) \left( \log(6(d_\mathcal{H} + d_\mathcal{T})) - 1 \right) + 2d_\mathcal{H} \log c \\
&\leq d/6 + (d_\mathcal{H} + d_\mathcal{T}) \log(6(d_\mathcal{H} + d_\mathcal{T})) + 2d_\mathcal{H} \log c
\end{aligned}$$

where the second step follows from the first-order Taylor series expansion of logarithm function at the point $6(d_\mathcal{H} + d_\mathcal{T})$. Therefore,

$$\begin{aligned}
d &\leq \frac{(d_\mathcal{H} + d_\mathcal{T}) \log(6(d_\mathcal{H} + d_\mathcal{T})) + 2d_\mathcal{H} \log c - d_\mathcal{T}(\log(d_\mathcal{T}))}{\log 2 - 1/6} \\
&\leq 2 \left( (d_\mathcal{H} + d_\mathcal{T}) \log(6(d_\mathcal{H} + d_\mathcal{T})) + 2d_\mathcal{H} \log c \right)
\end{aligned}$$

where the last step follows from $\log 2 - 1/6 < 1/2$. $\qquad\square$

## 1.3 Proof of Corollary 4.4

The first two conclusions of corollary 4.4 are straightforward. We prove the last statement:

**Example.** Suppose the instance is embedded in a vector space $\mathcal{X} = \mathbb{R}^p$. Consider the problem (Example 5.1.3 in [4]) of binary classification with uniform noise rate which is modeled as a Logistic regression: $\mathbb{P}(O \neq y | x, y) = S(w^\mathsf{T} x)$ where $S$ is the sigmoid function and $w$ is the parameter. Then the cross-entropy loss becomes: $-\mathbb{1}\{o \neq \widehat{y}\} \log(S(w^\mathsf{T} x)) - \mathbb{1}\{o = \widehat{y}\} \log(1 - S(w^\mathsf{T} x))$. We have $d_\mathcal{T} \leq 2p + 2$.

*Proof.* Given $2p + 3$ points in $\mathcal{X} \times \mathcal{Y} \times \mathcal{O}$, without loss of generality, suppose there are at least $p + 2$ points such that $o \neq y$. For these points, the value of annotation loss only depends on $S(w^\mathsf{T} x)$. For any $u \in \mathbb{R}$, the classifier

$$f_{h,T,u}(x, o) = \mathbb{1}\{\ell_\mathcal{O}(h(x), T, (x, o)) > u\} = \mathbb{1}\{\log(S(w^\mathsf{T} x)) < -u\}$$

is a linear classifier with decision boundary $w^\mathsf{T} x = e^{-u}$. Since the VC dimension of hyperplanes of dimension $p$ is $p + 1$, we know these linear classifiers cannot classify $p + 2$ points arbitrarily. Therefore, the original $2p + 3$ points cannot be classified arbitrarily, and we have $d_\mathcal{T} \leq 2p + 2$. $\quad\square$

## 1.4 Proof of Theorem 5.2

**Theorem** (Separation). For all $x \in \mathcal{X}$, we denote the *induced distribution families* by label $y_i$ as $\mathcal{D}_i(x) \overset{\text{def}}{=\joinrel=} \{(T(x))_i : T \in \mathcal{T}\} \subseteq \mathcal{D}_{\mathcal{O}}$ (recall that $(T(x))_i$ is the $i^{\text{th}}$ row of $T(x)$), and the set of all possible predictions of the label as $\mathcal{H}(x) \overset{\text{def}}{=\joinrel=} \{h(x) : h \in \mathcal{H}\} \subseteq \mathcal{Y}$. Suppose

$$\gamma \overset{\text{def}}{=\joinrel=} \inf_{(x,i,j):p(x,y_i)>0,j\neq i,y_j\in\mathcal{H}(x)} \mathrm{KL}(\mathcal{D}_i(x) \parallel \mathcal{D}_j(x)) > 0 \tag{11}$$

Then $\mathcal{H}$ is learnable from the observations of $(X, O)$ with $\eta \geq \gamma > 0$ via the ERM of cross-entropy loss. We call $\gamma$ the *separation degree*.

Moreover, if (11) is not satisfied, then there exists a sequence of transitions $\{T^{(k)}\}_k \in \mathcal{T}$ and distributions $\{D_X^{(k)}\}_k$ over $\mathcal{X}$ such that $\lim_k \eta^{(k)} = 0$ , where $\eta^{(k)}$ is defined the same as $\eta$ in [C3], with the expectation (in the definition of the risk functions) being taken according to $T^{(k)}$ and $D_X^{(k)}$.

*Proof.* Denote the cross-entropy of two distributions $D_1$ and $D_2$ as $H(D_1, D_2)$ and the entropy of a distribution $D$ as $H(D)$. Let $\ell_{\mathcal{O}}$ be the cross-entropy loss, for a fixed $x \in \mathcal{X}$ we have that

$$\begin{aligned}
&\mathbb{E}_o[\ell_{\mathcal{O}}(h(x), T, (x, o))] - \mathbb{E}_o[\ell_{\mathcal{O}}(h_0(x), T_0, (x, o))] \\
&= H((T_0(x))_{h_0(x)}, (T(x))_{h(x)}) - H((T_0(x))_{h_0(x)}, (T_0(x))_{h_0(x)}) \\
&= H((T_0(x))_{h_0(x)}, (T(x))_{h(x)}) - H((T_0(x))_{h_0(x)}) \\
&= \mathrm{KL}((T_0(x))_{h_0(x)} \parallel (T(x))_{h(x)})
\end{aligned}$$

If $h(x) \neq h_0(x)$, then by the separation condition we have that

$$\mathrm{KL}((T_0(x))_{h_0(x)} \parallel (T(x))_{h(x)}) \geq \gamma$$

Also, if $h(x) = h_0(x)$, we have

$$\mathrm{KL}((T_0(x))_{h_0(x)} \parallel (T(x))_{h_0(x)}) \geq \mathrm{KL}((T_0(x))_{h_0(x)} \parallel (T_0(x))_{h_0(x)}) = 0$$

Therefore, for a fixed $h \in \mathcal{H}$

$$\begin{aligned}
&R_{\mathcal{O}}(h \circ T) - \inf_{T \in \mathcal{T}} R_{\mathcal{O}}(h_0 \circ T) \\
&= \mathbb{E}_{x,o}[\ell_{\mathcal{O}}(h(x), T, (x, o))] - \mathbb{E}_{x,o}[\ell_{\mathcal{O}}(h_0(x), T_0, (x, o))] \\
&\geq \mathbb{P}(h(x) \neq h_0(x)) \inf_{T,h(x)\neq h_0(x)} (\mathbb{E}_o[\ell_{\mathcal{O}}(h(x), T, (x, o))] - \mathbb{E}_o[\ell_{\mathcal{O}}(h_0(x), T, (x, o))]) \\
&\quad + \mathbb{P}(h(x) = h_0(x)) \inf_{T,h(x)=h_0(x)} (\mathbb{E}_o[\ell_{\mathcal{O}}(h(x), T, (x, o))] - \mathbb{E}_o[\ell_{\mathcal{O}}(h_0(x), T, (x, o))]) \\
&\geq \mathbb{P}(h(x) \neq h_0(x)) \inf_{T,h(x)\neq h_0(x)} (\mathbb{E}_o[\ell_{\mathcal{O}}(h(x), T, (x, o))] - \mathbb{E}_o[\ell_{\mathcal{O}}(h_0(x), T, (x, o))]) \\
&= \mathbb{P}(h(x) \neq h_0(x)) \inf_{T,h(x)\neq h_0(x)} \mathrm{KL}((T_0(x))_{h_0(x)} \parallel (T(x))_{h(x)}) \\
&\geq \mathbb{P}(h(x) \neq h_0(x))\gamma \geq 0
\end{aligned}$$

This shows the consistency condition [C2]. Also, if $\mathbb{P}(h(x) \neq h_0(x)) > 0$, notice that $\mathbb{P}(h(x) \neq h_0(x)) = R(h)$, we have

$$\eta = \inf_{R(h)>0} \frac{R_{\mathcal{O}}(h \circ T) - \inf_{T \in \mathcal{T}} R_{\mathcal{O}}(h_0 \circ T)}{R(h)} \geq \frac{\gamma R(h)}{R(h)} = \gamma > 0$$

This shows the identifiability condition [C3].

Moreover, if the condition (11) is not satisfied, by definition we have

$$\begin{aligned}
\gamma &= \inf_{(x,i,j):p(x,y_i)>0,j\neq i,y_j\in\mathcal{H}(x)} \mathrm{KL}(\mathcal{D}_i(x) \parallel \mathcal{D}_j(x)) \\
&= \inf_{(x,i,j):p(x,y_i)>0,j\neq i,y_j\in\mathcal{H}(x),D_i\in\mathcal{D}_i(x),D_j\in\mathcal{D}_j(x)} \mathrm{KL}(D_i \parallel D_j) \\
&= 0
\end{aligned}$$

This condition implies that for any $k \in \mathbb{N}^+$, there exists a 5-tuple

$$\left(x^{(k)}, y_i^{(k)}, y_j^{(k)}, D_i^{(k)}(x^{(k)}), D_j^{(k)}(x^{(k)})\right) \in \mathcal{X} \times \mathcal{Y} \times \mathcal{Y} \times \mathcal{D}_{\mathcal{O}} \times \mathcal{D}_{\mathcal{O}}$$

such that

- $p(x, y_i^{(k)}) > 0$

- $y_i^{(k)} \neq y_j^{(k)}$

- There is a $h_- \in \mathcal{H}$ such that $h_-(x^{(k)}) = y_j^{(k)}$

- $\mathrm{KL}(D_i^{(k)}(x^{(k)}) \parallel D_j^{(k)}(x^{(k)})) < \frac{1}{k}$

Now, let $D_X^{(k)}$ be the point mass distribution with probability one to be $x^{(k)}$, i.e., $D_X^{(k)}(\{x^{(k)}\}) = 1$. Then, we have $h_0(x^{(k)}) = y_i^{(k)}$ since $h_0$ has zero classification error. Also, let $T_0^{(k)} \in \mathcal{T}$ be such that its $i^{\text{th}}$ row is $D_i^{(k)}$, and $T_-^{(k)} \in \mathcal{T}$ be such that its $j^{\text{th}}$ row is $D_j^{(k)}$. We have

$$
\begin{aligned}
\eta^{(k)} &= \inf_{h \in \mathcal{H}: R(h) > 0} \frac{R_{\mathcal{O}}(h \circ T) - \inf_{T \in \mathcal{T}} R_{\mathcal{O}}(h_0 \circ T)}{R(h)} \\
&= \inf_{h \in \mathcal{H}: R(h) > 0} \frac{R_{\mathcal{O}}(h \circ T) - R_{\mathcal{O}}(h_0 \circ T_0^{(k)})}{R(h)} \\
&\leq \frac{R_{\mathcal{O}}(h_-^{(k)} \circ T_-^{(k)}) - R_{\mathcal{O}}(h_0 \circ T_0^{(k)})}{R(h_-^{(k)})} \\
&\leq \mathrm{KL}(D_i^{(k)}(x^{(k)}) \parallel D_j^{(k)}(x^{(k)})) \\
&\leq \frac{1}{k}
\end{aligned}
$$

Let $k \to \infty$ and the desired result follows. $\qquad\square$

## 1.5 Proof of Proposition 5.5

**Proposition** (Concentration). A sufficient condition for (11) is that for every $1 \leq i \leq c$, there exists a set $S_i \subset \mathcal{O}$ (we call them *concentration sets*) such that

$$
\gamma_C \stackrel{\text{def}}{=\!=} \inf_{(i,j,x,T): T \in \mathcal{T}, p(x) > 0, j \neq i} \mathbb{P}_T(O \in S_i | x, y_i) - \mathbb{P}_T(O \in S_j | x, y_i) > 0 \tag{12}
$$

where $\mathbb{P}_T(\cdot)$ is the conditional probability defined by transition $T$. Under this condition, we can relate identifiability and separation degree by $\eta \geq \gamma \geq 2\gamma_C^2$. Since a condition imposed on all $T \in \mathcal{T}$ can be regarded as an assumption imposed on the true transition $T_0$, condition (12) can be rewritten as:

$$
\gamma_C = \inf_{(i,j): p(x, y_i) > 0, i \neq j} \mathbb{P}(O \in S_i | x, y_i) - \mathbb{P}(O \in S_j | x, y_i) > 0 \tag{13}
$$

Also, in this case, one can ensure learnability by the ERM which minimizes the following transition-independent annotation loss

$$
\ell_{\mathcal{O}}(h(x), T, (x, o)) \stackrel{\text{def}}{=\!=} \mathbb{1}\{o \notin S_{h(x)}\} \tag{14}
$$

For this annotation loss, we can bound the identifiability level by $\eta \geq \gamma_C$.

*Proof.* First, for any $(x, y_i) \in \mathcal{X} \times \mathcal{Y}$ with $p(x, y_i) > 0$ and $D_i \in \mathcal{D}_i(x), D_j \in \mathcal{D}_j(x)$, by Pinsker's inequality, we have

$$\text{KL}(D_i \parallel D_j) \geq 2\|D_i - D_j\|_{\text{TV}}^2 = \frac{1}{2}\|D_i - D_j\|_1^2$$

$$= \frac{1}{2}\left(\sum_{o \in \mathcal{O}} |D_i(o) - D_j(o)|\right)^2$$

$$\geq \frac{1}{2}(|D_i(S_i - S_j) - D_j(S_i - S_j)| + |D_i(S_j - S_i) - D_j(S_j - S_i)|)^2$$

$$\geq \frac{1}{2}(D_i(S_i - S_j) - D_j(S_i - S_j) - D_i(S_j - S_i) + D_j(S_j - S_i))^2$$

$$= \frac{1}{2}(D_i(S_i - S_j) - D_i(S_j - S_i) + D_j(S_j - S_i) - D_j(S_i - S_j))^2$$

$$= \frac{1}{2}(D_i(S_i) - D_i(S_j) + D_j(S_j) - D_j(S_i))^2$$

$$\geq \frac{1}{2}(2\gamma_C)^2 = 2\gamma_C^2$$

where $D_i(\cdot)$ is the probability measure over $\mathcal{O}$ defined by $D_i$, and $S_i - S_j$ is the set subtraction: $S_i - S_j \overset{\text{def}}{=\!=} \{o : o \in S_i \wedge o \notin S_j\} \subset \mathcal{O}$. Taking infimum on both sides of the inequality gives the first result. Another proof for this result can be found in the proof of Proposition 5.8.

Next, consider the annotation loss $\ell_{\mathcal{O}}(h(x), T, (x, o)) = \mathbb{1}\{o \notin S_{h(x)}\}$ and its ERM. Then we have

$$\mathbb{E}_{x,o}[\ell_{\mathcal{O}}(h(x), T, (x, o))] - \mathbb{E}_{x,o}[\ell_{\mathcal{O}}(h_0(x), T, (x, o))]$$

$$= \mathbb{P}(o \notin S_{h(x)}) - \mathbb{P}(o \notin S_{h_0(x)})$$

$$\geq \mathbb{P}(h(x) \neq h_0(x)) \inf_{x: h(x) \neq h_0(x)} \left(\mathbb{P}(o \in S_{h_0(x)}) - \mathbb{P}(o \in S_{h(x)})\right)$$

$$\geq \mathbb{P}(h(x) \neq h_0(x))\gamma_C = R(h)\gamma_C$$

Therefore,

$$\eta = \inf_{R(h)>0} \frac{R_{\mathcal{O}}(h \circ T) - \inf_{T \in \mathcal{T}} R_{\mathcal{O}}(h_0 \circ T)}{R(h)} \geq \frac{\gamma_C R(h)}{R(h)} = \gamma_C > 0$$

as claimed. $\square$

### 1.6 Proof of Proposition 5.8

**Proposition** (Evidence). A sufficient condition for (11) is that there exists Lipschitz (with respect to $L^1$-norm) functions $\Phi_{ij} : \mathbb{R}^c \to \mathbb{R}(1 \leq i \neq j \leq c)$ (we call them *evidence*) with Lipschitz constants $L_{ij}$ such that

$$\gamma_{ij} \overset{\text{def}}{=\!=} \inf_{p(x,y_i)>0, y_j \in \mathcal{H}(x), D_i \in \mathcal{D}_i(x), D_j \in \mathcal{D}_j(x)} \Phi_{ij}(D_i) - \Phi_{ij}(D_j) > 0 \qquad (15)$$

In this case, the separation degree can be bounded by $\gamma \geq \frac{1}{2}\min_{i \neq j}\left(\frac{\gamma_{ij}}{L_{ij}}\right)^2$.

*Proof.* Since $\Phi_{ij}$ is Lipschitz, then for any $a, b \in \mathbb{R}^s$, we have

$$|\Phi_{ij}(a) - \Phi_{ij}(b)| \leq L_{ij}\|a - b\|_1$$

Hence, given $(x, i, j)$ such that $p(x, y_i) > 0, j \neq i$ and $y_j \in \mathcal{H}(x)$, then for any $D_i \in \mathcal{D}_i(x)$ and $D_j \in \mathcal{D}_j(x)$, by Lipschitz property we have

$$\|D_i - D_j\|_1 \geq \frac{1}{L_{ij}}|\Phi_{ij}(D_i) - \Phi_{ij}(D_j)| \geq \frac{\gamma_{ij}}{L_{ij}}$$

Therefore, by Pinsker's inequality, we have

$$\text{KL}(D_i \parallel D_j) \geq \frac{1}{2}\|D_i - D_j\|_1^2 \geq \frac{1}{2}\left(\frac{\gamma_{ij}}{L_{ij}}\right)^2 \geq \frac{1}{2}\min_{i \neq j}\left(\frac{\gamma_{ij}}{L_{ij}}\right)^2$$

Taking infimum on the left hand side of the inequality gives the desired result.

In particular, if $\Phi$ represents the inner product with a fixed vector $u$, i.e., $\Phi(a) = \langle u, a \rangle$, then $\Phi$ is Lipschitz since for any $a, b \in \mathbb{R}^s$, by the Hölder's inequality, we have

$$|\Phi(a) - \Phi(b)| = |\langle u, a - b \rangle| \leq \|u\|_\infty \|a - b\|_1$$

Therefore, we can bound the Lipschitz constant of $\Phi$ by $L \leq \|u\|_\infty$.

To recover the concentration condition, given sets $S_i \subset \mathcal{O}(1 \leq i \leq c)$, for any $i \neq j$, let

$$\Phi_{ij}(a) = \left\langle \sum_{k:o_k \in S_i} \hat{e}_k - \sum_{k:o_k \in S_j} \hat{e}_k, a \right\rangle$$

Then $\Phi_{ij}(D_i) = \mathbb{P}_{D_i}(O \in S_i) - \mathbb{P}_{D_i}(O \in S_j)$ and $\Phi_{ij}(D_j) = \mathbb{P}_{D_j}(O \in S_i) - \mathbb{P}_{D_j}(O \in S_j)$. The concentration condition (13) implies that

$$\inf_{p(x,y_i)>0, y_j \in \mathcal{H}(x), D_i \in \mathcal{D}_i(x), D_j \in \mathcal{D}_j(x)} \Phi_{ij}(D_i) - \Phi_{ij}(D_j) \geq 2\gamma_C > 0$$

Moreover, since $\left\| \sum_{k:o_k \in S_i} \hat{e}_k - \sum_{k:o_k \in S_j} \hat{e}_k \right\|_\infty = 1$, the separation degree can be bounded by $\gamma \geq \frac{1}{2} \min_{i \neq j} (\gamma_{ij})^2 = 2\gamma_C^2$. $\qquad \square$

## 1.7   Proof of Proposition 5.10

**Proposition.**  Suppose the separation degrees of $y_i$ to $y_j$ of $O_1$ and $O_2$ are $\gamma_{i \to j1}$ and $\gamma_{i \to j2}$ respectively. Then, if the joint transition class is constructed as

$$\mathcal{D}_i(x) = \{\lambda D_1 + (1 - \lambda)D_2 : D_1 \in \mathcal{D}_{i1}(x), D_2 \in \mathcal{D}_{i2}(x)\}$$

then the separation degrees of $y_i$ to $y_j$ for the joint supervision satisfies:

$$\gamma_{i \to j} \leq \lambda \gamma_{i \to j1} + (1 - \lambda)\gamma_{i \to j2}$$

Also, if $\mathcal{O}_1 \cap \mathcal{O}_2 = \varnothing$, then the two sides are equal. As a consequence, a necessary condition for $y_i$ being separated from $y_j$ by the joint signal $O$ is that $y_i$ must be separated from $y_j$ by one of $O_1, O_2$.

*Proof.* Given $D_i \in \mathcal{D}_i(x)$ and $D_j \in \mathcal{D}_j(x)$, write $D_i = \lambda D_{i1} + (1 - \lambda)D_{i2}$ and $D_j = \lambda D_{j1} + (1 - \lambda)D_{j2}$, where $D_{i1} \in \mathcal{D}_{i1}(x)$, $D_{j1} \in \mathcal{D}_{j1}(x)$, $D_{i2} \in \mathcal{D}_{i2}(x)$, $D_{j2} \in \mathcal{D}_{j2}(x)$. The summation $D_i = \lambda D_{i1} + (1 - \lambda)D_{i2}$ means that we combine $D_{i1}$ and $D_{i2}$ as distributions over $\mathcal{O}$ such that $D_i(o) = \lambda \mathbb{1}\{o \in \mathcal{O}_1\}D_{i1}(o) + (1 - \lambda)\mathbb{1}\{o \in \mathcal{O}_2\}D_{i2}(o)$ for any $o \in \mathcal{O}$.

The first result basically follows from the convexity of KL-divergence: we have

$$\begin{aligned} \mathrm{KL}(D_i \parallel D_j) &= \mathrm{KL}(\lambda D_{i1} + (1 - \lambda)D_{i2} \parallel \lambda D_{j1} + (1 - \lambda)D_{j2}) \\ &\leq \lambda \mathrm{KL}(D_{i1} \parallel D_{j1}) + (1 - \lambda)\mathrm{KL}(D_{i2} \parallel D_{j2}) \end{aligned} \tag{16}$$

Hence,

$$\lambda \mathrm{KL}(D_{i1} \parallel D_{j1}) + (1 - \lambda)\mathrm{KL}(D_{i2} \parallel D_{j2}) \geq \inf_{x:p(x,y_i)>0, y_j \in \mathcal{H}(x)} \mathrm{KL}(\mathcal{D}_i \parallel \mathcal{D}_j) = \gamma_{i \to j}$$

Take infimum again on the left hand side of the inequality, we have

$$\lambda \gamma_{i \to j1} + (1 - \lambda)\gamma_{i \to j2} \geq \gamma_{i \to j}$$

More over, if $\mathcal{O}_1 \cap \mathcal{O}_2 = \varnothing$, then in (16), we have

$$\begin{aligned} &\mathrm{KL}(\lambda D_{i1} + (1 - \lambda)D_{i2} \parallel \lambda D_{j1} + (1 - \lambda)D_{j2}) \\ &= \sum_{o \in \mathcal{O}} (\lambda D_{i1}(o) + (1 - \lambda)D_{i2}(o)) \log\left( \frac{\lambda D_{i1}(o) + (1 - \lambda)D_{i2}(o)}{\lambda D_{j1}(o) + (1 - \lambda)D_{j2}(o)} \right) \\ &= \sum_{o \in \mathcal{O}_1} \lambda D_{i1}(o) \log\left( \frac{\lambda D_{i1}(o)}{\lambda D_{j1}(o)} \right) + \sum_{o \in \mathcal{O}_2} (1 - \lambda)D_{i2}(o) \log\left( \frac{(1 - \lambda)D_{i2}(o)}{(1 - \lambda)D_{j2}(o)} \right) \\ &= \lambda \mathrm{KL}(D_{i1} \parallel D_{j1}) + (1 - \lambda)\mathrm{KL}(D_{i2} \parallel D_{j2}) \end{aligned}$$

Hence taking infimum on both sides gives $\lambda \gamma_{i \to j1} + (1 - \lambda)\gamma_{i \to j2} = \gamma_{i \to j}$.

The above discussion shows that if $\gamma_{i \to j} > 0$, then one of $\gamma_{i \to j1}$ and $\gamma_{i \to j2}$ must be positive.

## 1.8 Remark 5.11

Finally, we show by a simple example that if $\mathcal{O}_1 \cap \mathcal{O}_2 \neq \varnothing$, then even if both $\lambda\gamma_{i\to j1}$ and $\gamma_{i\to j2}$ are positive, we can still have $\lambda\gamma_{i\to j} = 0$.

Consider a binary classification ($\mathcal{Y} = \{\pm 1\}$) with two noisy annotations (crowdsourcing with two annotators) $O_1$ and $O_2$. Suppose the transitions of the two annotations are known to the learner and are given by constant matrices

$$T_1(x) \equiv \begin{bmatrix} 0.6 & 0.4 \\ 0.4 & 0.6 \end{bmatrix} \text{ and } T_2(x) \equiv \begin{bmatrix} 0.4 & 0.6 \\ 0.6 & 0.4 \end{bmatrix}$$

Then, individually, both the annotations can ensure separation. However, suppose $\lambda = 1/2$, then in this case, if the annotations are mixed (i.e., the learner do not distinguish the annotations of different annotators, and hence $\mathcal{O} = \mathcal{O}_1 \cup \mathcal{O}_2 = \mathcal{Y}$), then for any $x, y$,

$$\mathbb{P}(O = y|x, y) = \lambda\mathbb{P}(O_1 = y|x, y) + (1 - \lambda)\mathbb{P}(O_2 = y|x, y) = 1/2$$

Here we used the condition that $\mathbb{1}\{O = O_k\}$ is independent with $X$. Now, it is not possible to learn $Y$ from the observation of $O$ since $O$ is simply a random noise that is independent of $Y$. $\qquad\square$