[Reviews · NeurIPS 2020]

Review 1

Summary and Contributions: The paper studies the problem of learning when direct annotations are not directly accessible, and rather a stochastic function of the real annotation is used as the supervision signal. Knowledge of the exact transition function from the real label to the noisy observation is not assumed. The paper provides a unified framework to study identifiability and learnability of the problem using and generalizing tools from PAC theory.

Strengths: The problem studied in the paper is general and important both from a theoretical perspective and in practice in many applications of machine learning. The paper is well-written, and the proposed theory is interesting and clearly explained.

Weaknesses: My main concern with the paper is about the use of VC-dimension style arguments and the lack of any empirical validation. Regarding the former: The paper introduces the problem of indirect supervision using the example of a labeling problem. However, in most real applications the hypothesis class H has a very large VC-dimension and typically leads to very vacuous bounds. Could the authors discuss how this affects their theory? While it seems the question of whether the system is identifiable should not be affected by this, does this theory inform us, at least qualitatively, of the number of additional samples required to learn the system using partial observation? On a related topic, in the case of complex classification systems like DNNs non-vacuous bounds have been obtained by using PAC-Bayes bounds rather than PAC bounds. Is a similar theory to the one proposed applicable to that style of bounds? Regarding empirical validation, I understand that the scope of the paper is mainly to build a theoretical framework for learnability with partial annotations. However, since the theoretical bounds such as eq. (2) may be vacuous in practice and even if the system is learnable, the number of samples required could be unrealistic in practice. For these reasons, I would have expected experiments to show whether the number of additional annotations required to learn when using superset annotation, noisy labels, or different annotators (Examples 5.6, 5.7, 5.12) is well predicted (at least qualitatively) in a practical setup. Even a simple experiment with CIFAR-10 classification and simulated noise in the labels would do for this. Lacking experiments, I would have liked to see a more extensive discussion of what are the implications of the framework for current practice.

Correctness: I have only skimmed through the proofs, but the high-level intuitions seem correct. There is no empirical verification of the theory.

Clarity: The paper is well written, all required concepts are clearly introduced and the reasonings are well explained and easy to follow.

Relation to Prior Work: Prior work is discussed, although not extensively. I would have appreciated a section ion the appendix with a more extensive comparison of these known results mentioned in Section 2.

Reproducibility: Yes

Additional Feedback:


Review 2

Summary and Contributions: 1. The paper analyses the problem of learning from indirect labels when the true labels are realizable. 2. The authors introduce a general formalism using transition kernels that describes how indirect labels may arise, which includes several standard examples as special cases. 3. The authors introduce a notion of complexity for collections of (classifier, transition kernel) pairs that allow them to derive upper bounds on the *true* risk of the learned hypothesis. Their result depends on a *consistency* assumption, which can be interpreted as saying that the true risk minimizer for the indirect labels is the true risk minimizer for the true labels; and depends quantitatively on an *identifiability* condition. 4. They present a generic way to learn from indirect labels using the cross entropy loss, and characterize the consistency and identifiability conditions for the cross entropy loss in terms of the KL divergence between sets of distributions which might arise from different true labels for each instance.

Strengths: 1. The results appear to be correct. 2. The results are quite interesting and were not obvious. I think this is a useful and novel contribution. 3. The algorithm needed to achieve their results is implementable; minimizing the cross entropy is something that could be done numerically. 4. This is relevant to NeuRIPS, and it provides a theoretical justification for methods that are already used in many applications. 5. The fact that indirect labels are widely used means that this work can have a high impact and that the theoretical foundations in this area are important to practitioners as well.

Weaknesses: 1. Are there lower bounds to match the main upper bound for the main result, Thm.~4.2, Eq.~2? If there was only a single T which was the identity then "no" because you can get the faster rate from realizable PAC learning. How would this need to be generalized to have matching upper and lower bounds? At least a discussion of the matter would help make the limitations of the present work more clear. 2. I think the work would benefit by having a concrete example of the phenomenon described in line 199-202. E.g. if there are only 2 indirect labels and 4 real labels, show a concrete example of learning the true labelling funciton for the more complex case.... Maybe just have linear 1-D thresholds with fixed distances between the transitions discontinuity in classes, with noisy subsetting. Both working through this analytically, to give intuition for why we can learn more labels with less labels is possible, and some empirical results would be useful. POST FEEDBACK COMMENTS: Regarding R3's comments on the realizablility assumption being restrictive, and the author's response: I agree that this assumption is quite restrictive and unrealistic in practice. I believe that DNNs are a bad example, considering that the authors require the "true" hypothesis to belong to a VC class (of low VC dimension relative to sample size) in their main theorem. This is because neural networks notoriously have VC dimension far too large for any results based upon their VC dimension to be relevant in practice. Furthermore, it is well known (Zhang et al. '16, Nagarajan & Kolter '19, etc) that an attempt to apply uniform convergence directly to DNNs will lead to vacuous bounds, and the proof techniques of the present work rely on uniform convergence. Regardless of the choice of example to justify it, the work would be more compelling if the realizability assumption was not required. However, I agree with the part of the authors' response to that question that says what they have is something to build upon rather than a final solution to the problem of learning with indirect labels. Perhaps a discussion of the issues that led to the authors needing the realizability assumption would be reasonable for an appendix, and could lead to subsequent work that removes this assumption?

Correctness: The results all appear to be correct to me.

Clarity: 1. The paper is mostly well written. 2. there are a few places where the writing was awkward or had grammatical mistakes a (non-comprehesive list) of the ones i had marked as I read... - in the abstract: "when, often," - line 63: "This two papers" - line 191 : "moreove" 3. Some notation was confusing; -e.g. $E_{X,Y}[\ell(h(x),y)]$; should commit to upper or lower case letters for random variables, and the other for deterministic variabels - I did not understand what was meant by (lines 87, 88) "when using $y_i$ and $o_i$ as subscripts, we regard...."

Relation to Prior Work: 1. I think the literature review appears comprehensive. 2. I am not very familiar with the broader literature on learning with indirect labels, so I am not an appropriate reviewer to judge this. POST FEEDBACK COMMENTS: Regarding R3's comments on related work: I had to read the Ben-David & Schuller and Ben-David and Borbely papers since I was not familiar with them. I believe that the similarities between the two pieces of work are limited, and I would *not* say that the results of present paper are a direct consequence of the results of that other work. The fact that the transformations are not deterministic, and only post transformation samples are available are significant differences that make the present work interesting and novel. The deterministic and invertible transformations of the Ben-David paper necessarily do not lose or distort any information from the original training point, while applying a non-trivial Markov kernel to the data does lose information. Maybe the most important difference, as mentioned in their response, is that even if one could manipulate the spaces involved so that problem analyzed by the present work was a special case of the previous work mentioned, the types of sample complexities are materially different. The present work does not need any pre-tansformation samples to learn. Indeed, with access to the number of pre- transformation samples as would be required by the other papers, the problem in the present paper would be rendered essentially trivial. However, since the present work has access to zero pre- transformation examples it is a significantly different problem, at least in my opinion. While it would be reasonable to cite these two papers, failing to do so alone should not be grounds for rejection, since in my opinion there are material differences in the problems addressed by the two pieces of work.

Reproducibility: Yes

Additional Feedback: 1. (Lines 107-109) The cross entropy is described as a marginal distribution but then defined with a conditional distribution. 2. Learnability (as defined around line 114) is a property of both the hypothesis set AND the collection transition kernels. The same $\mathcal{H}$ may be learnable for one $\mathcal{T}$ may not be learnable for another. The phrasing should include $\mathcal{T}$ for this reason... e.g "$\mathcal{H}$ is indirectly learnable with respect to \mathcal{T}" or something like that 3. In Theorem 4.2, remind the reader what $b$ is. Is there a good reason to not just assume that $b$ is 1? 4. The Natarajan dimension is not defined in the present work. It would make the paper easier to read if it was, since it is referenced in line 151. 5. Line 152 should say $\ell_\mathcal{O}\circ \mathcal{T}$ instead of $\ell\circ \mathcal{T}$ I htink 6. The definition of $\mathcal{D}_i(x)$ is confusing. Can this be restated in a different way so that it is more clear what set of measures this is? Maybe not using the subscript shortcut that had a definition that confused me? 7. Line 202: I think \dim should be cardinality, since these are finite sets. 8. Eq.~(11) shows how to combine different indirect labelling mechanism by mixing. This corresponds to either getting a label of type 1 xor a label of type 2; which I will call a "direct sum" label. In the crowdsourcing labels example, this is like asking user 1 or user 2 but not both to label a given point. It would be useful to characterize "product" labels as well, where you get both a label of type 1 and a label of type 2, equivalently the indirect label space is $\mathcal{O}_1 \time \mathcal{O}_2$ and the kernel space is given by products of the two types of kernels. Is it possible to characterize the separation of the product indirect label? Naively its at least as good as the better of the 2 I think? maybe even bigger than the min_{i,}(max_{u\in\{1,2\}} \gamma_{i\to j} under indirect label scheme u)? Whatever the good lower bound is, this gives you an interesting calculus of indirect labeling mechanisms which would allow you to answer questions about more complex mixed label schemes, like what the separation is when you have 10 users and a random subset of 5 of them label each example. 9. An intuitive explanation of conditions [C2] and [C3] would be useful. I would describe [C2] as "The true labeling function is also the best under the O-risk on the indirect labels" or something like that. Not sure how to intuitively describe [C3]. 10. My score (7) is based on the present condition of the manuscript. If the authors addressed all of my feedback adequately in the rebuttal or a revision then I would likely raise my score to an 8. To raise my score above an 8. the authors would need to show me they have made an additional important contribution that I have failed to acknowledge and that would increase the impact of the paper over what I have understood, or that I have underestimated the importance of some contribution of the paper. POST FEEDBACK COMMENTS: Overall, based on the authors' response, I stand by my original judgment of "7: A good submission; accept." I have no issues with their responses to my points, however as other reviewers have pointed out that the realizability assumption is quite restrictive, and I do not feel like the authors have addressed that concern as well as they could have I will not be increasing my score.


Review 3

Summary and Contributions: The paper analyzes the sample complexity of learning a multiclass classifier from imperfectly labeled training samples. It provides upper bounds on that sample complexity that are based on some quantification of the prior knwledge about the correlations between the labels provided by the imperfect supervision and the correct labels and the complexity of the target labeling rule.

Strengths: The paper addresses a natrally arising problem, is clearly written and mathematically sound.

Weaknesses: The paper relies on rather well known tools for analyzing sample complelxity and makes rather unrealistically strong assumptions (such as the realizability assumption - that the target labeling rule belongs to a low capacity class of labveling rules known to the learner, the assumption that the the "noise" of the training labeling belongs to a small capacity known class of transitions, and the "indentifiability" assumption. Furthermore, a rather similar analysis has been earlier published and this submission seems ignorant of that. The paper "Exploiting task relatedness for multiple task learning" by Ben-David and Schuller in Learning Theory and Kernel Machines, 567-580 (COLT 2003). See also "A notion of task relatedness yielding provable multiple-task learning guarantees" S Ben-David, RS Borbely Machine learning 73 (3), 273-287. That old paper applies a rather similar framework and analysis for a slightly different problem - multiple task learning. The main difference being that there the imprfect labels are due to differences between tasks, rather than to some noising process.

Correctness: The results seem correct.

Clarity: Yes, it is, except fo rits ignorance of closely related previous work (as detailed above).

Relation to Prior Work: No. My main issue with this paper is that it ignores closely related previous work (as detailed above) and to a certain extent rediscovers ideas discused in that paper.

Reproducibility: No

Additional Feedback:


Review 4

Summary and Contributions: The submission proposes a theoretical analysis of weakly-supervised learning. The analysis uses a very general formulation of weakly-supervised learning, which covers a series of supervision forms including superset learning, label corruption, and joint supervision. The analysis also covers a few previous results as special cases. The analysis is solid, and the writing is clear.

Strengths: The analysis is very general, which covers multiple supervision forms. The bounds in the analysis seems to be tight. The weak VC-major is clever: it avoids tedious dimensions for different types of problems. The result is very relevant to the community.

Weaknesses: I have read the paper again, checked other reviews, and also the response. I have identified the following weakness in this submission: 1. The two papers "Exploiting task relatedness for multiple task learning" and "A notion of task relatedness yielding provable multiple-task learning guarantees" should be analyzed in this submission. Previous results in these papers should be included in section 4. The results in these two papers should directly apply to some cases of weak supervisions. 2. The realizable case is a weakness of the paper, and the response to this question is not persuasive -- deep models probably cannot fit into VC analysis in this submission. The analysis of this submission is more focused on weak supervision, so the analysis should not be bonded to realizable case. The submission really needs to give an outlook about how to carry such analysis to non-realizable cases. Base on others' opinions, I am downgrading my rating to 7.

Correctness: I have read through the paper and proofs and find no critical flaws.

Clarity: The paper is well written. Interpretations of some key results can be improved. In (4) and (5), can you explain why gamma_C = 0 makes the problem not learnable anymore? 2. line 191: “moreove” -> moreover

Relation to Prior Work: Prior work are discussed in details.

Reproducibility: Yes

Additional Feedback:

[Author Response · NeurIPS 2020]

We thank all the reviewers for their insightful comments and suggestions that will all be incorporated in the final version.
We start our response with Reviewer #3, since some of the comments made there are inaccurate.

**Response to Reviewer #3**: The major critique is that our paper rediscovers prior ideas. However, the two papers of
Ben-David and others pointed out by the reviewer are essentially different in the setups and results, which is also why
those papers were not cited in more related works we cite (e.g., [11,17,31]) either. Below we expand these differences:

1. The setting. They assumed a *multitask learning* setting where $n$ tasks/datasets symmetrically contribute to the loss
function, and the learner has access to *datasets for all tasks* (i.e., theorem 3.5 assumes $|S_i| \geq m$ for all $i$). In contrast,
our framework assumes an asymmetric *supervision* setting where the learner only has access to samples of $(X, O)$
but *no dataset for the target label $Y$*. Moreover, They assumed that all tasks are *binary classification* while in our
framework $\mathcal{Y}$ and $\mathcal{O}$ can be *multiclass* and different. Also, they define the relatedness of tasks by deterministic
*functions $X \to X$* while in we define it as *conditional distributions $\mathbb{P}(O|Y)$*.
2. The analysis and results. In their work, since all tasks are associated with a dataset of size $\geq m$, the generalization is
given for-free, as long as the generalized VC-dimension is finite. However, in our paper, learnability is nontrivial
(e.g., think about a very noisy dataset). So, our paper makes additional effort to characterize the transition class and
the learner's prior knowledge about it, which results in conditions [C2] and [C3].

The other critique is we make strong assumptions. In fact, our assumptions are *common* and often *weaker* than literature.

1. Realizability assumption. This is commonly assumed in the related works (see line 34) and is approximately
satisfied by many practical models (such as DNNs that achieve classification error close to 0 on many large, practical
problems). Due to the stochastic generating process of the indirect signal, it would be technically hard to remove
this assumption in a general setting. We therefore think that it is a good starting point for developing the theory.
2. Learnability conditions. First, the conditions [C2] and [C3] actually generalize and relax many standard learning
conditions in the literature (see Example 5.6 & 5.7). Moreover, the second part of our theorem 5.2 shows that the
conditions are not only sufficient, but also close to be necessary. In other words, our framework provides more
practical (rather than unrealistic) ways to determine learnability and/or find learnable supervision signals.

**Response to Reviewer #1**: Here are our responses to the concerns:

1. Use of VC-style argument. We choose to use VC-style argument to bound the Rademacher complexity because it is
a *safe* way to show *learnability*. It is safe in the sense that most of the practical machine learning models, including
DNNs, have a finite VC-dimension. Also, it is possible to adopt to a PAC-bayesian argument by defining induced
prior and posterior via the transitions. However, to show learnability in this way, we would need to further bound the
KL-divergence by constructing priors for specific models. It would be an interesting direction for future research.
2. Practical implications. We will address this issue more clearly in the final version. Briefly: To make full use of the
indirect observations, our framework provides a practical and unfied way to understand the supervision power of the
signal when only weaker form of the prior knowledge is given (encoded by *separation*). Our framework generalizes
previous results, and hence suggests new learning scenarios. Also, in a non-learnable case, section 5.3 also guides to
find complementary signals to make the problem learnable.

**Response to Reviewer #2**: We thank the reviewer for the useful feedback. For the weakness and additional feedback:

1. To derive a matching lower bound, we will need two things: (1) For the case when $\mathcal{T} = \{I\}$, it is true that faster
rate can be achieved. This is partly because the task $X \to O$ is agnostic in default. So if using the current method,
(at least) we will need to define what is a "realizable" $O$. (2) A new measure similar to separation that links the
annotation and classification risk. We will add more discussion on this issue and/or the limitations fo current work.
2. We will add an illustrating example for this phenomenon 199-202. For linear case, the intuition is that different
regions will have different patterns of the distribution of $O$. We will also add a visualization to make it clear.
3. For the additional feedback: **(1,5)** There are typos. Will be corrected in the complete version. **(2)** We will adopt
the suggested way of definition and call it "$\mathcal{T}$-learnable". **(3)** $b$ is the upper bound of the annotation loss $\ell_{\mathcal{O}}$, which
may not be the 0-1 loss. We will add a reminder here. **(4)** We will add a definition for the Natarajan dimension.
**(6)** We can restate the definition as: $\mathcal{D}_i(x)$ is the set of all the induced distributions $\mathbb{P}_T(O|X = x, Y = y_i)$. **(7)**
$\dim$ is used to indicate the singularity of $T$, since one may concern if a singular $T$ will lead to information loss. **(8)**
Our framework can be extended to product case and the product label will contain more information and provably
have better supervision power. However, to formulate this idea in general, we also need to model and characterize
the *correlations* between different annotations (which could be expensive). It will be an interesting future research
direction. **(9)** Your intuition is correct. We would describe it like this: [C2]: the optimal classifier of $Y$ can induce
an optimal predictor of $O$. [C3]: the suboptimal classifier of $Y$ will induce a suboptimal predictor of $O$.

[Meta-Review · NeurIPS 2020]

This paper considers the theoretical learnability of a multi-class classifier when supervision comes in the form of a variable that has nonzero mutual information with the true label. The reviewers agree this is an important problem and the paper makes useful, sound contributions, including the notion of separation to characterize generalization bounds. The major concern was the relationship between this work and earlier work, particularly Ben-David and Schuller (COLT '03), which uses a notion of transition functions F that are seemingly similar to this work's transformation functions. After the author response and discussion, the reviewers identified several key distinctions in this work. These distinctions include that the transformations are non-deterministic, that the pre- and post-transformation label cardinalities can be different, and that only the post-transformation samples are available for training. We suggest that the reviewers clarify these differences in the camera ready version. (We also note that R3 indicated in discussion that they would like to raise their score to a 6, but did not officially update it.)